# Janus particle-engineered structural lipiodol droplets for arterial embolization

Sijian Tao[1,2,7], Bingquan Lin[3,7], Houwang Zhou[1], Suinan Sha[1], Xiangrong Hao[1], Xuejiao Wang[1], Jianping Chen[1], Yangning Zhang[1], Jiahao Pan[1], Jiabin Xu[4], Junling Zeng[5], Ying Wang[1], Xiaofeng He[4], Jiahao Huang ®[2,6] ✉, Wei Zhao ®[4] ✉ & Jun-Bing Fan ®[1] ✉

Embolization (utilizing embolic materials to block blood vessels) has been considered one of the most promising strategies for clinical disease treatments. However, the existing embolic materials have poor embolization effectiveness, posing a great challenge to highly efficient embolization. In this study, we construct Janus particle-engineered structural lipiodol droplets by programming the self-assembly of Janus particles at the lipiodol-water interface. As a result, we achieve highly efficient renal embolization in rabbits. The obtained structural lipiodol droplets exhibit excellent mechanical stability and viscoelasticity, enabling them to closely pack together to efficiently embolize the feeding artery. They also feature good viscoelastic deformation capacities and can travel distally to embolize finer vasculatures down to 40 μm. After 14 days post-embolization, the Janus particle-engineered structural lipiodol droplets achieve efficient embolization without evidence of recanalization or non-target embolization, exhibiting embolization effectiveness superior to the clinical lipiodol-based emulsion. Our strategy provides an alternative approach to large-scale fabricate embolic materials for highly efficient embolization and exhibits good potential for clinical applications.

Embolization is a minimally invasive process for delivering embolic materials to block blood vessels and has been considered one of the first-line therapy strategies for treating diseases[1–3], such as hepatocellular tumors, gastrointestinal tract bleeding, and renal tumors[4–6]. A variety of embolic materials (represented by solid beads and liquid embolic materials) have been widely used in clinical practice[5,7–9], especially in tumor chemoembolization. However, owing to the different sizes (from 5–10 μm to 1–2 cm in diameter) and complex architectures of blood vessels, most embolic materials remain limited by efficient chemoembolization[5,10]. In the arterial chemoembolization

of tumors, such as transarterial chemoembolization (TACE), beyond embolization of large-sized arterial vasculatures, the embolic materials are also expected to travel distally to embolize those finer arterial vasculatures to achieve distal embolization, which is beneficial to enhance the efficiency of chemoembolization[11–13]. However, the solid beads with sizes >100 μm are usually limited by embolizing the finer arterial vasculatures probably because of their poor viscoelastic deformation abilities[13–16]. Liquid embolic materials, such as lipiodol, are promising to flow to finer vasculatures and show a good capacity for radiography and drug loading, which has attracted particular

[1]Cancer Research Institute, School of Basic Medical Sciences, Southern Medical University, 510515 Guangzhou, P. R. China. [2]School of Biomedical Engineering, Southern Medical University, 510515 Guangzhou, P. R. China. [3]Department of Medical Imaging Center, Nanfang Hospital, Southern Medical University, 510515 Guangzhou, P. R. China. [4]Division of Vascular and Interventional Radiology, Department of General Surgery, Nanfang Hospital, Southern Medical University, 510515 Guangzhou, P. R. China. [5]Laboratory Animal Research Center of Nanfang Hospital, Southern Medical University, 510515 Guangzhou, P. R. China. [6]Department of Critical Care Medicine, Affiliated Hospital of Guangdong Medical University, 524000 Zhanjiang, P. R. China. [7]These authors contributed equally: Sijian Tao, Bingquan Lin. ✉e-mail: jhuangaf@connect.ust.hk; pummpa@smu.edu.cn; fjb2012@mail.ipc.ac.cn

attention for tumor chemoembolization over the past several decades[17,18]. However, these lipiodol systems are highly unstable in the blood and remain limited by, e.g., their recanalization, non-specific embolization, toxicity, etc[19–22]. Therefore, there is a critical need for new embolic materials that can adapt to the architectures of the blood vessels for efficient arterial embolization.

Janus particles feature two distinct physical or chemical properties on their surfaces, like the two-faced Roman God Janus. They have attracted significant attention in the context of emulsion stabilization owing to their excellent interfacial activities in comparison with their isotropic particles and molecular surfactants[23,24]. Amphiphilic Janus particles combining the advantages of the high desorption energy of spherical particles and the amphiphilicity of molecular surfactants can more efficiently adsorb and self-assemble onto interfaces to stabilize droplets[25–29]. We recently demonstrated a general emulsion interfacial polymerization approach for enabling the large-scale synthesis of amphiphilic Janus particles with tunable topologies and surface chemistries[30,31]. With this in mind, we propose that these amphiphilic Janus particles may provide an opportunity to develop new embolic materials by segregating the lipiodol–water interface (so as to fabricate stable droplets). The self-assembly of Janus particles at the lipiodol–water interface enables the pure lipiodol to form stable and viscoelastic droplets[32–34]. Compared to lipiodol, the self-assembly of the Janus particles at the lipiodol–water interface endows the obtained structural lipiodol droplets with good mechanical stability, allowing them to closely pack together to embolize a feeding artery. Compared to rigid solid beads, the self-assembly of Janus particles at the lipiodol–water interface may also endow the obtained structural lipiodol droplets with good viscoelasticity, allowing them to travel distally to embolize finer artery vasculatures via deformation.

Herein, we discuss a conceptually distinct design based on programming the self-assembly of amphiphilic Janus particles at the lipiodol–water interface to fabricate structural lipiodol droplets. Based on the droplets, we achieve highly efficient renal embolization in rabbits. The obtained Janus particle-engineered structural lipiodol droplets exhibit excellent mechanical stability and a capacity for viscoelastic deformation. As expected, they can change into different shapes from their original spherical shapes under external forces, such as into ellipsoidal-shaped, dumbbell-shaped, and snowman-shaped structures. The excellent viscoelastic deformation capacity of the Janus particle-engineered structural lipiodol droplets allows them to adapt to different levels of blood vessels for efficient embolization. Therefore, these Janus particle-engineered structural lipiodol droplets can facilitate efficient embolizations of blood vessels (Fig. 1).

## Results

### Large-scale fabrication of Janus particle-engineered structural lipiodol droplets

In this study, the amphiphilic Janus particles were fabricated based on an emulsion interfacial polymerization strategy[30] (The detailed fabrication can be found in the "Methods" section). Scanning electron microscope (SEM) images of the Janus particles showed crescent shapes with sizes of $2.26 \pm 0.15\,\mu m$ and surface potentials of approximately $-42.5\,mV$ (Supplementary Fig. 1). The obtained amphiphilic Janus particles comprised poly(styrene-co-divinyl benzene) (PSDVB) and poly(acrylic acid) (PAA). As indicated in our previous reports, the concave surface comprised hydrophobic PSDVB and the convex surface comprised hydrophilic PAA[30,31]. We used these amphiphilic Janus particles to fabricate the Janus particle-engineered structural lipiodol droplets. As a typical synthesis, when 0.1 mL of lipiodol were mixed with 6 mL of an aqueous solution containing Janus particles (1.5 mg/mL) under shearing, the Janus particles could rapidly self-assemble at the lipiodol–water interface to fabricate Janus particle-engineered structural lipiodol droplets (Fig. 2a). In contrast, it was difficult to fabricate stable lipiodol droplets by using spherical polystyrene-PAA

particles (Supplementary Fig. 2). The obtained Janus particle-engineered structural lipiodol droplets exhibited spherical shapes with sizes of $120 \pm 40\,\mu m$ (Fig. 2b, c). We have calculated that 1 mL lipiodol can produce 6000 Janus particle-engineered structural lipiodol droplets ($120 \pm 40\,\mu m$). Magnifying the surface of a Janus particle-engineered structural lipiodol droplet demonstrated that a large number of Janus particles were tightly attached to the interface of the obtained lipiodol droplet (Fig. 2d–f). This suggested that the amphiphilic Janus particles could easily adsorb and self-assemble onto the surface of the lipiodol droplet to form a densely segregating layer and stabilize the droplet. Computed tomography (CT) images confirmed the excellent radiographic capacity of the obtained Janus particle-engineered structural lipiodol droplets; in contrast, clinical 8spheres® beads had no radiographic capacity (Fig. 2g and Supplementary Fig. 3). To characterize the arrangements of the Janus particles onto the interface of lipiodol droplet, the Janus particles and lipiodol were labeled with amino fluorescein dye and Nile red, respectively. The results clearly demonstrated that the Janus particles were densely distributed onto the interface of the lipiodol droplet; in addition, the overlap between the Janus particles (green fluorescence) and lipiodol (red fluorescence) exhibited a yellow color (Fig. 2h). The sizes of Janus particle-engineered structural lipiodol droplets could be controlled from $25 \pm 16$ to $480 \pm 53\,\mu m$ by tuning the concentration of Janus particles and rates of lipiodol and deionized water (Fig. 2i, j and Supplementary Figs. 4 and 5). We used the Janus particle-engineered structural lipiodol droplets with sizes of $120 \pm 40\,\mu m$ to perform all the subsequent experiments.

Next, the cytotoxicity, stability, and large-scale production of the Janus particle-engineered structural lipiodol droplets were comprehensively investigated. The cytotoxicities of the Janus particles, lipiodol, and Janus particle-engineered structural lipiodol droplets were respectively evaluated by co-incubating them with human umbilical vein endothelial cells (HUVECs). The results suggested that the Janus particles, Janus particle-engineered structural lipiodol droplets, and lipiodol exhibited significantly low cytotoxicity, even at concentrations as high as $32\,\mu g/mL$ (Supplementary Fig. 6a). Moreover, we used cisplatin to test the drug-loading capacity of our Janus particle-engineered structural lipiodol droplets; the results suggested that the encapsulation efficiency reached >90% at 6 mg and 10 mg of cisplatin, as detected by an inductively coupled plasma optical emission spectrometer (Supplementary Table 1); in other words, for example, in a typical encapsulation of cisplatin at 6 mg, 1 mg of cisplatin could be loaded into approximately 107 Janus particle-engineered structural lipiodol droplets in the case. When the Janus particle-engineered structural lipiodol droplets were loaded with cisplatin, their cytotoxicity to HepG2 cells significantly increased (Supplementary Fig. 6b). We performed the release profile of the Janus particle-engineered structural lipiodol droplets. The results demonstrated that the obtained cisplatin-loaded Janus particle-engineered structural lipiodol droplets exhibited pH-responsive drug release due to the carboxyl group of Janus particles (Supplementary Fig. 7). The long-term structural stability of the Janus particle-engineered structural lipiodol droplets was assessed for 12 months. The results indicated that the sizes of the Janus particle-engineered structural lipiodol droplets did not change significantly (Supplementary Fig. 8). When the fabrication was scaled up to 200 times that of the aforementioned feed, this approach produced approximately 25 g of Janus particle-engineered structural lipiodol droplets in one batch, providing an effective way to produce the structural lipiodol droplets at a large scale. The CT images also showed the excellent radiopacity of these Janus particle-engineered structural lipiodol droplets (Supplementary Fig. 9). In addition, the Janus particle-engineered structural lipiodol droplets were easily packaged (Supplementary Fig. 10), indicating their good potential for future clinical application. Hemocompatibility is a critical characteristic to investigate any newly developed embolic materials in directly

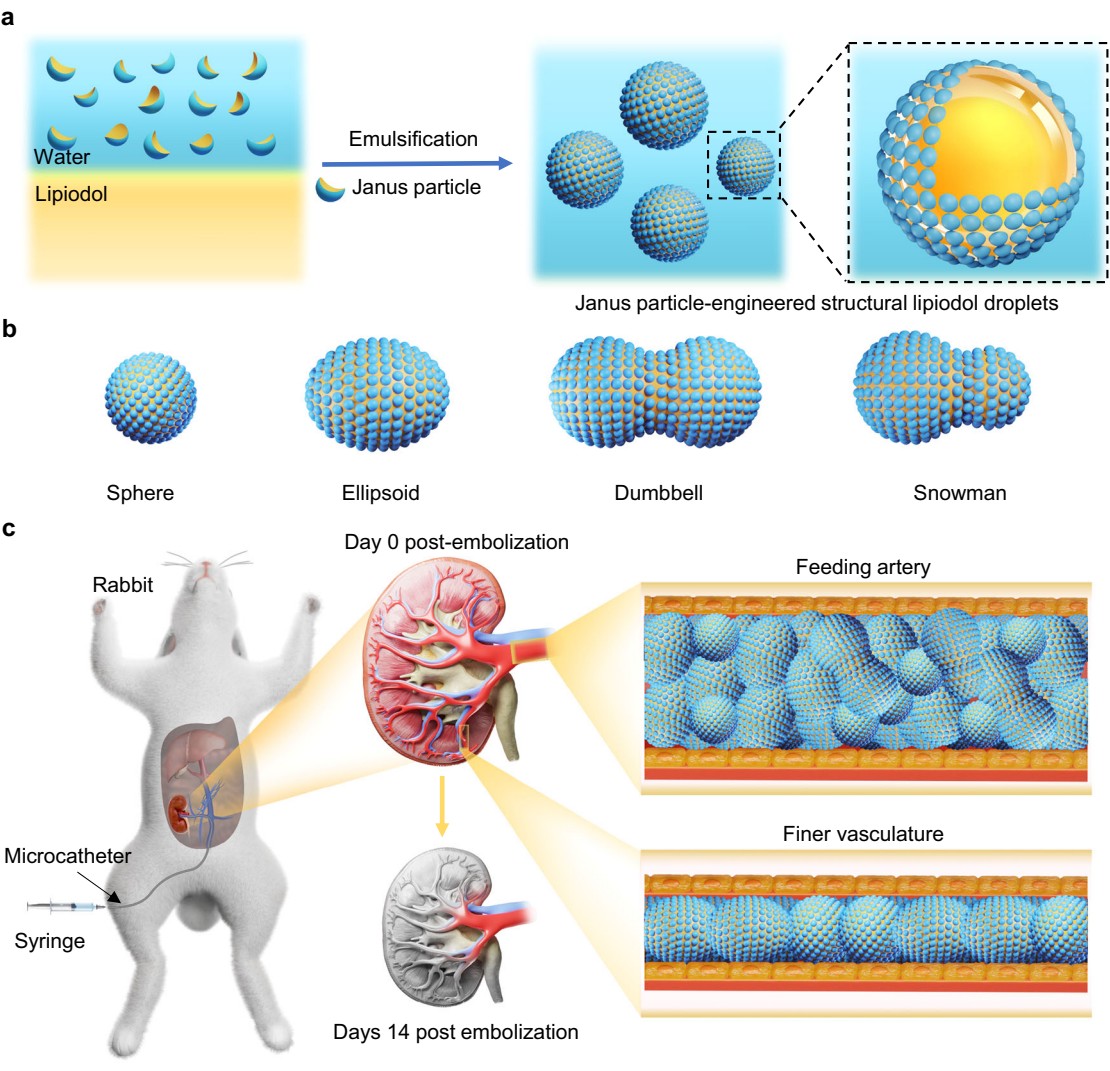

**Fig. 1 | Janus particle-engineered structural lipiodol droplets for arterial embolization. a** Schematic representation of the fabrication of Janus particle-engineered structural lipiodol droplets. When the Janus particles' aqueous solution was sheared with lipiodol, the particles could rapidly self-assemble at the lipiodol–water interface to fabricate Janus particle-engineered structural lipiodol droplets. **b** Schematic representation of the shape evolution of Janus particle-engineered structural lipiodol droplets. The obtained Janus particle-engineered structural lipiodol droplets featured an excellent capacity for viscoelastic deformation and could further evolve into different shapes under external forces, such as ellipsoidal-shaped, dumbbell-shaped, and snowman-shaped structures. **c** Janus particle-engineered structural lipiodol droplets for renal embolization in a rabbit. During the process of visible embolization in the rabbit, Janus particle-engineered structural lipiodol droplets adapted to different levels of the blood vessels to persistently embolize the feeding arteries and traveled distally to embolize finer vasculatures via viscoelastic deformation.

contacting with blood. We investigated the hemocompatibility of the Janus particle-engineered structural lipiodol droplets by hemolysis assays. The results indicated that the overall hemolysis rate of the Janus particle-engineered structural lipiodol droplets with different sizes was <2.5% (a hemolysis rate <5% is considered permissible), suggesting that the Janus particle-engineered structural lipiodol droplets could not trigger hemolysis (Supplementary Fig. 11). These results demonstrated that the obtained Janus particle-engineered structural lipiodol droplets with excellent radiopacity, stability, and tunable size as well as hemocompatibility could be successfully fabricated by programming the self-assembly of Janus particles at the lipiodol–water interface.

### Viscoelasticity of Janus particle-engineered structural lipiodol droplets and their embolization effectiveness in vitro

We subsequently investigated the viscoelastic deformation capacity of the obtained Janus particle-engineered structural lipiodol droplets. We

found that the obtained Janus particle-engineered structural lipiodol droplets had excellent viscoelastic deformation capacities under external pressure. We then performed rheological experiments to evaluate the viscoelastic characteristics of the lipiodol droplets. As shown in Fig. 3a, the viscosity of the Janus particle-engineered structural lipiodol droplets and clinical 8spheres® beads significantly decreased with an increase in the shear rate, thereby exhibiting good shear-thinning behavior. The results suggested that the decreased viscosity of these Janus particle-engineered structural lipiodol droplets and clinical 8spheres® beads upon shearing was beneficial to delivery. Moreover, we also performed injectability tests of Janus particle-engineered structural lipiodol droplets with different sizes. The result demonstrated that the breakloose force and injection force of all the Janus particle-engineered structural lipiodol droplets was <10 N, suggesting good transcatheter injection (Fig. 3b). In addition, the scanning oscillation amplitude results showed that the maximum storage modulus (G′) of the Janus particle-engineered structural lipiodol

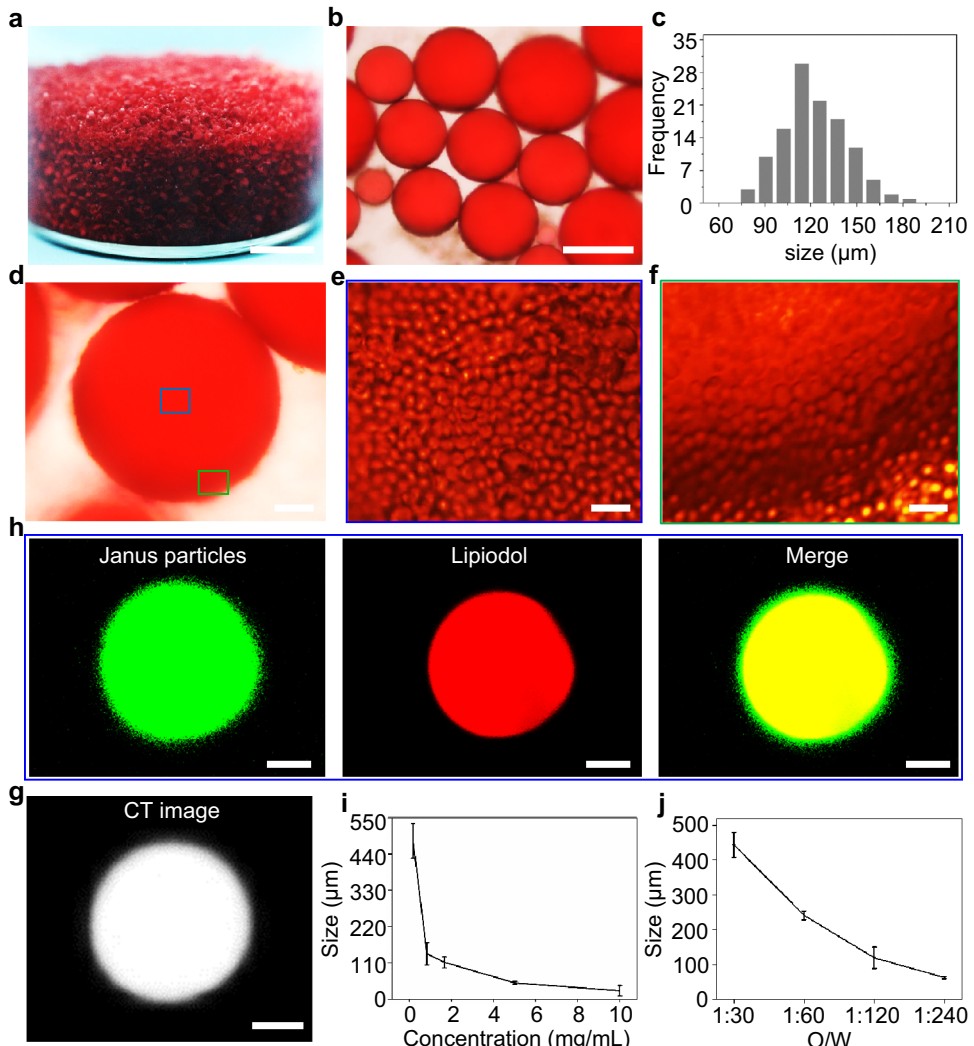

**Fig. 2 | Large-scale fabrication of Janus particle-engineered structural lipiodol droplets. a** Photograph of the obtained Janus particle-engineered structural lipiodol droplets. Scale bars: 3 mm. **b** Optical microscope images of the obtained Janus particle-engineered structural lipiodol droplets. Scale bars: 100 μm. **c** Size distribution of Janus particle-engineered structural lipiodol droplets. The obtained Janus particle-engineered structural lipiodol droplets exhibited spherical shapes with sizes of 120 ± 40 μm. **d**–**f** Optical microscope images of the distribution of Janus particles on the interface of a lipiodol droplet. Scale bars: 20 μm. **g** Computed tomography (CT) image of the obtained Janus particle-engineered structural lipiodol droplet. Scale bars: 40 μm. **h** Confocal microscopy images of a single Janus particle-engineered structural lipiodol droplet. The Janus particles and lipiodol

were labeled with amino fluorescein dye and Nile red, respectively. The results clearly demonstrated that these Janus particles were densely distributed onto the interface of a lipiodol droplet; the overlap between the Janus particles (green fluorescence) and lipiodol (red fluorescence) appeared yellow. Scale bars: 40 μm. **i, j** Size control of Janus particle-engineered structural lipiodol droplets. The sizes of Janus particle-engineered structural lipiodol droplets could be controlled from 25 ± 16 to 480 ± 53 μm by tuning the concentration of Janus particles and rates of lipiodol and deionized water (n = 3 independent samples). Data are presented as means ± SD. Experiments were performed three times (**a**, **b**, **d**–**f**, **h**), with similar results.

droplets was approximately 710 Pa, while the maximum G′ of clinical 8spheres® beads was approximately 3900 Pa (Fig. 3c). The results indicated that Janus particle-engineered structural lipiodol droplets exhibited much more viscoelastic deformation capacities than the clinical 8spheres® beads. In contrast, the clinical lipiodol exhibited liquid characteristics (Fig. 3c). The Janus particle-engineered structural lipiodol droplets (with excellent radiopacity) could be deformed into ellipsoid-shaped, dumbbell-shaped, and snowman-shaped structures under external stress (Fig. 3d, e). Meanwhile, we also dynamically observed the Janus particle-engineered structural lipiodol droplets passing through a finer glass capillary tube. As shown in Supplementary Fig. 12, they passed through the finer glass capillary tube through viscoelastic deformation.

Next, a glass capillary tube was used to visually mimic the embolization of the obtained Janus particle-engineered structural

lipiodol droplets. The right side of the capillary glass tube was sealed. The Janus particle-engineered structural lipiodol droplets were injected from the left side to the right side of the capillary glass tube at a constant speed of 0.5 mL/min. When those Janus particle-engineered structural lipiodol droplets preferentially reached the end of the tube (right side), they began to pack together tightly. Some of them that approached the right side of the capillary glass tube were deformed into different shapes under continuous injection force (Fig. 3f). To further verify the packing capacity, we compared the packing density of our Janus particle-engineered structural lipiodol droplets with that of clinical 8spheres® beads. The packing density was calculated as follows[35–37]:

$$D_{rcp}(R,H) = D_{rcp}(\infty) + A/R + B/H \qquad (1)$$

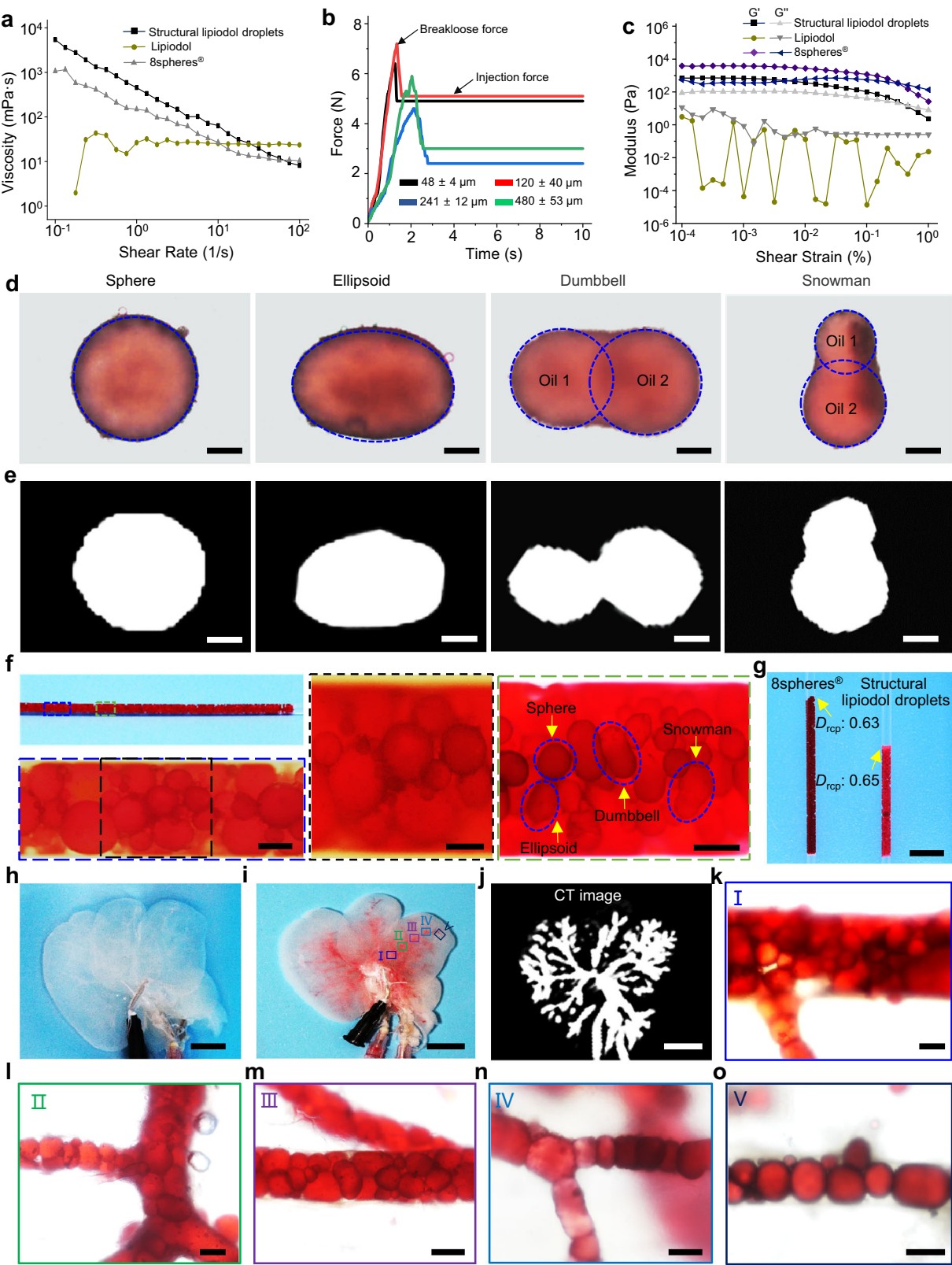

where $D_{rcp}$ ($R$, $H$) is the packing density of spheres in a given cylinder with radius $R$ and height $H$. $D_{rcp}$ ($\infty$) is the desired density of infinite dimensions, and $A$ and $B$ are constants[35]. We filled capillary glass tubes with the same numbers and sizes of clinical 8spheres® beads and Janus particle-engineered structural lipiodol droplets, respectively. As shown in Fig. 3g, from Eq. (1), compared to the clinical 8spheres® beads (packing height of 2.2 cm), the Janus particle-engineered structural lipiodol droplets (packing height of 1.3 cm) exhibited a more efficient packing capacity. It was calculated that the clinical 8spheres® beads reached a packing density of approximately 0.63, whereas the Janus particle-engineered structural lipiodol droplets reached a packing density of approximately 0.65. This result indicated that our Janus particle-engineered structural lipiodol droplets could pack more densely than the clinical 8spheres® beads, making them

**Fig. 3 | Viscoelasticity and embolization effectiveness of Janus particle-engineered structural lipiodol droplets in vitro. a** Rheology of the Janus particle-engineered structural lipiodol droplets, lipiodol and 8spheres® beads by shear rate sweeps. The viscosity of the Janus particle-engineered structural lipiodol droplets and the 8spheres® beads significantly decreased with the increase of the shear rate, which is beneficial to transcatheter injection. **b** The injection force curves of Janus particle-engineered structural lipiodol droplets with different sizes. **c** Storage modulus (G′) and loss modulus (G″) of the Janus particle-engineered structural lipiodol droplets, lipiodol, and 8spheres® beads as characterized by oscillatory strain sweeps, respectively. **d** Optical microscope images of Janus particle-engineered structural lipiodol droplets with different shapes. Scale bars: 30 μm. **e** CT images of Janus particle-engineered structural lipiodol droplets with different shapes. Scale bars: 30 μm. **f** Embolization effectiveness of Janus particle-engineered structural lipiodol droplets in a capillary glass tube. Scale bars: left is 100 μm, center is 100 μm, and right is 200 μm. **g** Comparison of packing densities of clinical 8spheres® beads and Janus particle-engineered structural lipiodol droplets. Scale bars: 0.5 cm. **h** Photograph of the decellularized liver of a Sprague-Dawley rat. Scale bar: 1 cm. **i** Photograph of the decellularized liver after injecting Janus particle-engineered structural lipiodol droplets into blood vessels. Scale bar: 1 cm. **j** CT images of the decellularized liver after embolizing of Janus particle-engineered structural lipiodol droplets. The Janus particle-engineered structural lipiodol droplets smoothly distributed into the whole blood vessels. Scale bar: 1 cm. **k–o** Optical microscope images of the embolization effectiveness of Janus particle-engineered structural lipiodol droplets within different levels of blood vessels of decellularized liver. Within these Figures, I, II, III, IV, V represents the different blood vessel zone in decellularized liver from (**h**). Janus particle-engineered structural lipiodol droplets are closely packed into the blood vessels. Except for embolizing the feeding artery, they could also travel distally to embolize finer vasculatures by viscoelastic deformation. Scale bars: 100 μm. Experiments were performed three times (**d–o**), with similar results. Source data are provided as a Source Data file.

beneficial for enhancing the embolization effectiveness in blood vessels.

Next, we used the decellularized liver of a Sprague-Dawley (SD) rat to further assess the embolization effectiveness of the obtained Janus particle-engineered structural lipiodol droplets in vitro. The results demonstrated that the droplets could be smoothly delivered into the whole blood vessels (Supplementary Fig. 13). When 1 mL of the Janus particle-engineered structural lipiodol droplets was slowly injected into the decellularized liver, they could be closely packed into the blood vessels (Fig. 3h–j). Except for embolizing the feeding artery, the obtained Janus particle-engineered structural droplets with excellent viscoelastic deformation capacity could also travel distally to embolize finer vasculatures by deformation (Fig. 3k–o). These results demonstrated that our Janus particle-engineered structural lipiodol droplets could adapt to the different levels of blood vessels; this would be expected to result in highly efficient embolization.

## Embolization effectiveness of Janus particle-engineered structural lipiodol droplets in the kidneys of rabbits

To further evaluate the embolization effectiveness of the obtained Janus particle-engineered structural lipiodol droplets in vivo, a renal embolization was performed in rabbits. To facilitate comparison, we only embolized the right kidney. A 2.2 F microcatheter was used to catheterize the right renal arteries from femoral artery access, and digitally subtracted angiography (DSA) was performed to monitor the blood vessels. Before and after embolization, 3 mL of contrast agent (iohexol) was injected into the blood vessels for the DSA evaluation. Subsequently, starting from a distal renal artery, 1.5 mL of embolic materials, including Janus particle-engineered structural lipiodol droplets (containing 1 mL lipiodol, approximately 6000 droplets), clinical 8spheres® beads (approximately 30,000 beads), and lipiodol-based emulsion (containing 1 mL lipiodol), were injected into the right kidney of rabbits via the 2.2 F coaxial catheter, respectively. The fluoroscopy results indicated that the Janus particle-engineered structural lipiodol droplets could be smoothly delivered and continually distributed into the blood vessels from the renal artery to the vascular branches (Supplementary Fig. 14 and Supplementary Movie 1). However, when we injected the clinical lipiodol-based emulsion, we could clearly observe leakage of lipiodol into the renal vein and subsequently into the lung, demonstrating a potential risk of pulmonary embolization (Supplementary Fig. 15 and Supplementary Movie 2). In clinical practice, 8spheres® beads have no radiopacity capacity, so they are usually additionally introduced with a contrast agent (such as iohexol) for DSA evaluations (Supplementary Fig. 16). In contrast, our Janus particle-engineered structural lipiodol droplets could be monitored in real-time during the entire process of embolization. On day 0 post-embolization, the DSA results indicated that the blood vessels could hardly be observed in the cases administered with Janus particle-engineered structural lipiodol droplets and clinical 8spheres® beads, suggesting

their good embolization capacities (Fig. 4a). After 14 days post-embolization, the Janus particle-engineered structural lipiodol droplets allowed for more efficient embolization, especially in the renal cortex, whereas the lipiodol-based emulsion group exhibited evident recanalization in most blood vessels. Moreover, the clinical 8spheres® beads also demonstrated recanalization in partial arteries near the renal hilum (Fig. 4a). CT images showed that the Janus particle-engineered structural lipiodol droplets remained visible in the blood vessels of the renal cortex after 14 days post-embolization and that the embolized right kidney could be three-dimensionally reconstructed based on the CT images. However, in the lipiodol-based emulsion group, most of lipiodol droplets had been metabolized, leading to a low CT intensity after 14 days post-embolization (Fig. 4b, c). The reduction rates for the embolized right kidney in the Janus particle-engineered structural lipiodol droplets group, clinical 8spheres® beads group, and clinical lipiodol-based emulsion group were calculated by AW VolumeShare 4.7 (reformat, volume, GE Healthcare) software as 38.85%, 38.56%, and 8.06%, respectively (Fig. 4d). Subsequently, the kidneys in each group were dissected and cut from the middle after 14 days post-embolization (Fig. 4e and Supplementary Fig. 17). The results demonstrated that, compared to the left kidney (Control), the volume of the embolized right kidney obviously reduced in the Janus particle-engineered structural lipiodol droplets and clinical 8spheres® bead groups. The embolized right kidney exhibited a milky white appearance, especially in the Janus particle-engineered structural lipiodol droplet group, suggesting an anemic infarction owing to blocking the supply of oxygen and nutrients during the process of embolization[38,39]. In the Janus particle-engineered structural lipiodol droplet group, the renal cortex of the embolized right kidney was nearly necrotic, and there was no significant recanalization. In contrast, in the lipiodol-based emulsion group, the volume of the embolized right kidney had no significant reduction and remained blood-red, indicating an evident recanalization. In the 8spheres® bead group, we also found that few blood vessels of the embolized right kidney had refilled with blood, suggesting recanalization in these blood vessels (Fig. 4e). Moreover, we monitored the other organs by CT images 14 days after embolization, and the results demonstrated no evidence of non-target embolization in the lung, liver, heart, spleen, brain, and normal left kidney in the Janus particle-engineered structural lipiodol droplets group (Fig. 4f). To investigate the systemic side effects after the embolization by the Janus particle-engineered structural lipiodol droplets, blood samples of rabbits were collected from each group before and after embolization on days 0, 3, 9, and 14. Routine blood examinations, liver and kidney function examinations, blood coagulation examinations, and weight tests were conducted, and the results showed that all of the values were within normal ranges (Supplementary Fig. 18). We also measured the inflammatory factors in the blood of rabbit post-embolization and the results suggested that the inflammatory factors in the blood of rabbit stayed within the normal

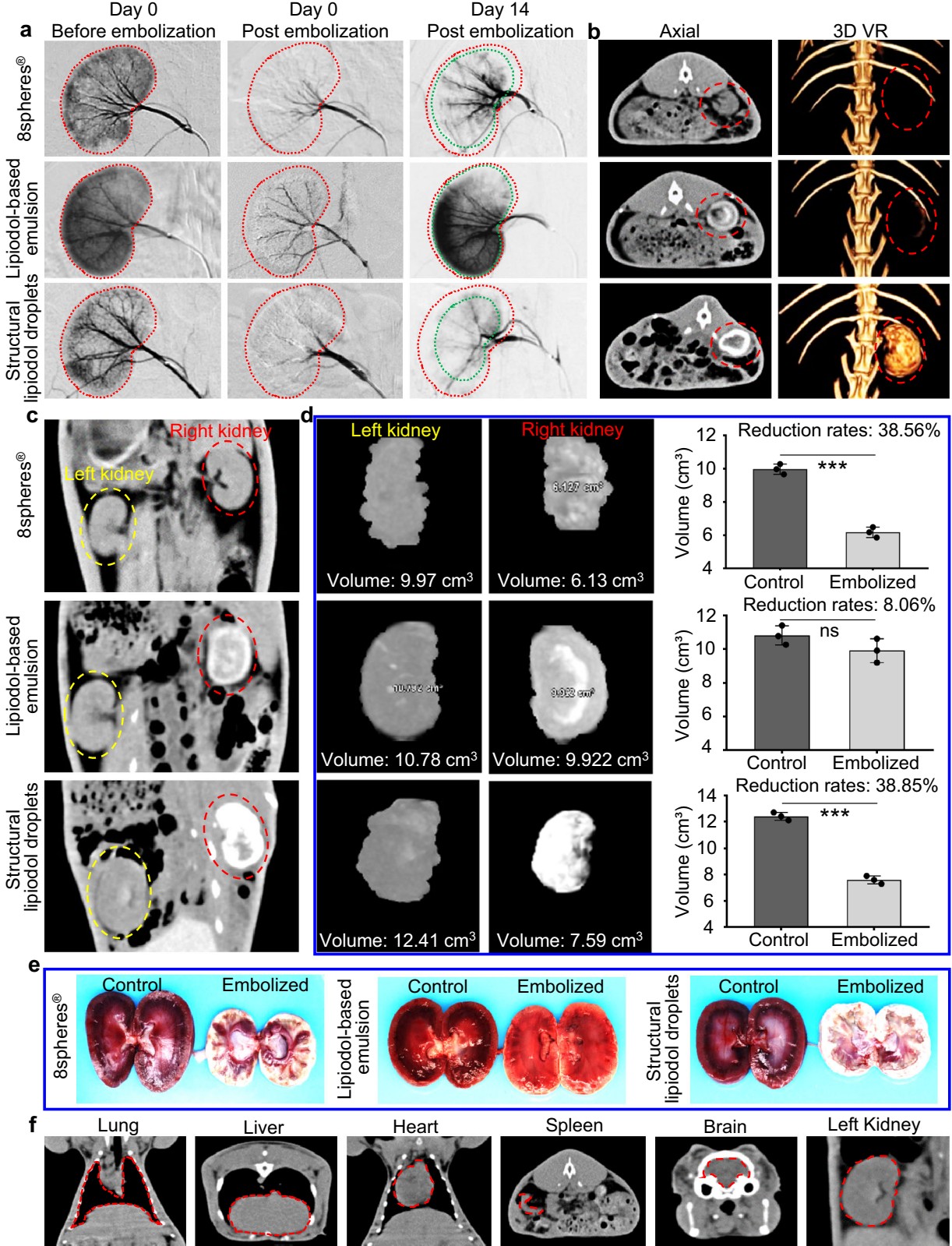

range in the Janus particle-engineered structural lipiodol droplet group (Supplementary Fig. 19).

Pathological analysis was performed after 14 days post-embolization to comprehensively verify the embolization effectiveness of the Janus particle-engineered structural lipiodol droplets, lipiodol-based emulsion, and 8spheres® groups, respectively. The results indicated that in the Janus particle-engineered structural

lipiodol droplets group, the infarcted area was mainly located between the renal cortex and medulla. Most of the renal cortex was necrotic, consistent with the characteristics of ischemic infarction. In the clinical 8spheres® bead group, there were few hyperemic zones, indicating that some blood vessels were recanalized in the embolized right kidney. This was consistent with the results from the DSA on day 14 post-embolization. Meanwhile, we also found that few normal glomerulus

**Fig. 4 | Embolization effectiveness of Janus particle-engineered structural lipiodol droplets in the kidneys of rabbits. a** Digitally subtracted angiography (DSA) of the right kidney before and after embolization on day 0 and 14 in different treatment groups. After 14 days post-embolization, the DSA results indicated that the Janus particle-engineered structural lipiodol droplets resulted in more efficient embolization (especially in the renal cortex), whereas the lipiodol-based emulsion group exhibited evident recanalization in most of the blood vessels. Additionally, the clinical 8spheres® beads also encountered recanalization in the partial arteries near the renal hilum. **b** CT images and three-dimensional virtual reality (3D VR) of the embolized right kidney. **c** CT images of the comparison of a normal left kidney and an embolized right kidney. On day 14 post-embolization, the Janus particle-engineered structural lipiodol droplets remained visible in the renal cortex. However, in the lipiodol-based emulsion group, most of lipiodol droplets had been metabolized, leading to a low CT intensity after 14 days post-embolization.

**d** Statistics of the volume of the left kidney (Control) and embolized right kidney in different groups after 14 days post-embolization by CT. The reduction rates were calculated by AW VolumeShare 4.7 (reformat, volume, GE Healthcare) software. Data are presented as mean values ± SD ($n = 3$ biologically independent experiments). Significant differences were evaluated by two-tailed unpaired $t$ test. \*\*\*$p < 0.001$ and ns ($p = 0.16$). **e** Photograph images of excised kidneys after 14 days post-embolization. Compared to the left kidney (Control), the embolized right kidney appeared milky white, and its volume was significantly reduced in the Janus particle-engineered structural lipiodol droplet group and 8spheres® bead group. **f** CT images of other organs after 14 days post-embolization. The results suggested no evidence of non-target embolization in the Janus particle-engineered structural lipiodol droplet group. Experiments were performed three times (**a**–**f**), with similar results.

remained in the 8spheres® bead group, whereas only necrotic glomerulus was observed in the Janus particle-engineered structural lipiodol droplets group (Fig. 5a). Besides, we also investigated additional tissue response in the embolized right kidney post-embolization. The results indicated that inflammation response and fibrosis in the microenvironment could be clearly observed in the Janus particle-engineered structural lipiodol droplets group and 8spheres® bead group (Supplementary Fig. 20). In particular, in the Janus particle-engineered structural lipiodol droplets group, an obvious infarction-triggered inflammation zone could be observed, into which a large number of immune cells were recruited. Moreover, we also investigated other major organs (heart, liver, spleen, lung, and left kidney) post-embolization in the Janus particle-engineered lipiodol droplet group. The results suggested that there was no obvious tissue damage and any inflammation or fibrosis in these organs (Supplementary Fig. 21). These results demonstrated that highly efficient embolization could be achieved using the Janus particle-engineered structural lipiodol droplets without any evidence of recanalization and non-target embolization. Furthermore, histological analysis of frozen sections of the embolized right kidney after 14 days post-embolization showed that the blood vessels were effectively embolized by our Janus particle-engineered structural lipiodol droplets (Supplementary Fig. 22). Hematoxylin and eosin (H&E) image results indicated that the Janus particle-engineered structural lipiodol droplets were closely packed into the blood vessels, even for the finer vasculatures; in this context, their deformation profiles could be observed (Fig. 5b and Supplementary Fig. 23). Based on the pathology results from the embolized blood vessels, we found that the Janus particle-engineered structural lipiodol droplets could embolize finer blood vessels down to 40 µm (Fig. 5c). Confocal microscopy and SEM images also confirmed a large number of Janus particles visible within the blood vessels (Fig. 5d, e). Moreover, these results also demonstrated that microemboli did not occur. In contrast, the 8spheres® beads were mainly observed in those large-sized arterial vasculatures and we did not observe them in the finer vasculatures (Supplementary Fig. 24). In the lipiodol-based emulsion group, the embolized right kidney exhibited poor efficiency and most regions remained maintained normal tissue structures after 14 days post-embolization (Fig. 5a and Supplementary Fig. 25).

**The fate of the Janus particle-engineered structural lipiodol droplets post-embolization in vivo**

Next, we investigated the short-term and long-term fate of the Janus particle-engineered structural lipiodol droplets post-embolization. The results indicated that accompanying with the volume reduction of kidney, the Janus particle-engineered structural lipiodol droplets could be gradually metabolized, regardless of Janus particles or lipiodol. As shown in Supplementary Fig. 26, the lipiodol intensity was gradually decreased with the prolonging of embolization time, indicating that the lipiodol within the Janus particle-engineered structural lipiodol

droplets could be slowly metabolized. To further explore the metabolic pathway of lipiodol within the Janus particle-engineered structural lipiodol droplets, we monitored the different organs by CT images on day 0, day 14, day 30, and day 45 post-embolization. The results demonstrated that except for kidney and bladder, no evidence of lipiodol distribution was observed in the lung, liver, heart, spleen, brain, and normal left kidney, suggesting that lipiodol was mainly metabolized via kidney (Supplementary Fig. 27).

We also investigated the metabolic pathway of Janus particles within the Janus particle-engineered structural lipiodol droplets. To monitor the fate of these non-degradable Janus particles, they were chemically modified with Cy7 fluorescent dyes. And then, these fluorescent Janus particles were used to fabricate Janus particle-engineered structural lipiodol droplets. To assess the biodistributions of Janus particles within the Janus particle-engineered structural lipiodol droplets, rabbits were sacrificed on day 45 post-embolization and the main organs were carefully excised for fluorescence imaging using an in vivo imaging system (IVIS Lumina II, Caliper, USA). The results indicated that except for kidney, the fluorescence of Janus particles within the Janus particle-engineered structural lipiodol droplets was also observed in the liver and bladder on 45 days post-embolization (Supplementary Fig. 28). And, in the collected urine of rabbits, we also clearly observed the Janus particles (Supplementary Fig. 29). These results indicated the metabolic pathway of Janus particles mainly proceeded with the liver and kidney.

## Discussion

Since 1978, Professor Yamada proposed the transcatheter arterial chemoembolization technique, and lipiodol-based embolization has been widely used in the clinic. In our study, lipiodol was formulated into droplets in a stable manner by programming the self-assembly of Janus particles at the lipiodol–water interface. In our current study, although efficient embolization was achieved by these Janus particle-engineered structural lipiodol droplets, their wide size distribution (120 ± 40 µm) may lead to difficulties in particle selection and administration with a lack of prediction for embolic efficacy. To address this issue, we have attempted to produce uniform Janus particle-engineered structural lipiodol droplets by microfluidic technique. In a typical fabrication, when a lipiodol droplet was extruded from a dispersion phase channel, the Janus particles in a continuous phase channel can self-assemble onto the interface of lipiodol droplets to fabricate Janus particle-engineered structural lipiodol droplets with uniform size distribution (127 ± 7 µm) (Supplementary Fig. 30).

In summary, we demonstrated a conceptually distinct design by programming the self-assembly of Janus particles at a lipiodol–water interface to fabricate Janus particle-engineered structural lipiodol droplets on a large scale for vascular embolization. The Janus particle-engineered structural lipiodol droplets

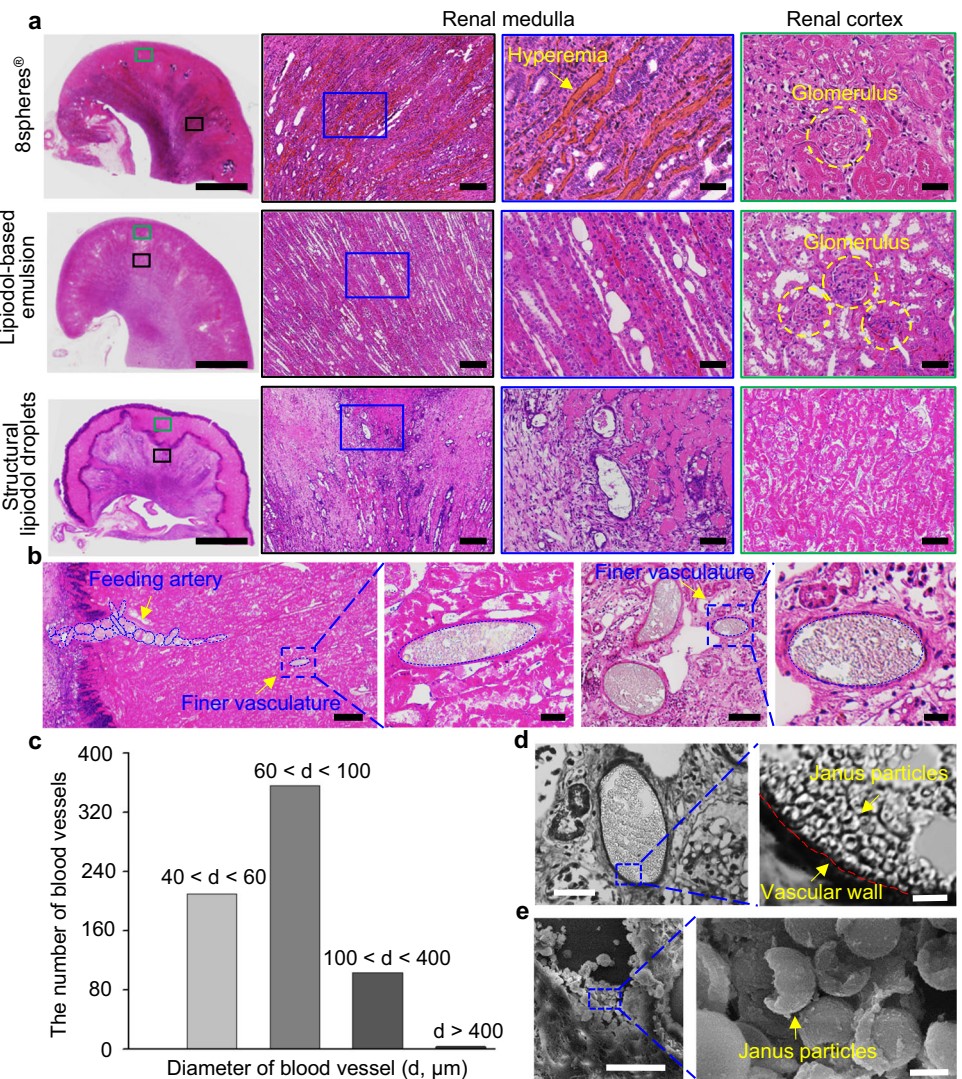

**Fig. 5 | Pathological analysis of the embolization effectiveness of Janus particle-engineered structural lipiodol droplets, lipiodol-based emulsion, and 8spheres® beads after 14 days post-embolization. a** H&E images of the embolized right kidney. The results indicated that in the Janus particle-engineered structural lipiodol droplet group, most of the renal cortex was necrotic. In the clinical 8spheres® bead group, there were few hyperemic zones, indicating that some blood vessels were recanalized in the embolized right kidney. In the lipiodol-based emulsion group, the embolized right kidney exhibited poor efficiency and most regions remained maintained normal tissue structures. Meanwhile, we also found that few normal glomeruli remained in the 8spheres® bead group, whereas only necrotic glomeruli were observed in the Janus particle-engineered structural lipiodol droplets group. The scale bars are 1000, 100, 100, and 50 μm from left to right. **b** H&E images of the embolized right kidney in the Janus particle-engineered structural lipiodol droplet group. Beyond artery embolization, these Janus particle-engineered structural lipiodol droplets could be delivered to finer vasculatures, with their viscoelastic deformation profiles clearly visible. The scale bars are 300, 30, 100, and 20 μm from left to right. **c** The number of embolized vessels in half of the right kidney according to pathological section results. **d, e** Confocal microscope and scanning electron microscope images of blood vessels in the embolized right kidney. The results clearly showed a large number of Janus particles within the blood vessels, even in finer vasculatures. Scale bars: left, 50 μm and right, 5 μm (**d**). Scale bars: left, 30 μm and right, 1 μm (**e**). Experiments were performed three times (**a, b, d, e**), with similar results.

exhibited excellent viscoelasticity and could further evolve into different shapes under external forces, exhibiting good flexibility for embolizing different levels of blood vessels. During the process of visible embolization in rabbits, they could adapt to different levels of blood vessels to persistently embolize the feeding arteries and travel distally to embolize finer vasculatures via viscoelastic deformation. As expected, highly efficient embolization was achieved by our Janus particle-engineered structural lipiodol droplets without any evidence of recanalization or non-target embolization. We believe that our Janus particle-engineered structural lipiodol droplets show potential as a new type of embolic material and can widen the scope of application by combining with therapeutic agents for the treatment of vessel-related diseases, especially in TACE treatment of tumors.

## Methods

### Cell lines and animals

Hepatocellular carcinoma (HepG2) cells were kindly provided by Prof. Yi Gao from The Second Clinical Medical College of Southern Medical University and which were cultured in high glucose DMEM complete medium (Biological Industries (BI)). HUVECs were obtained from the Cell Bank of Shanghai institute of Cell Biology, Chinese Academy of Sciences (Shanghai China) and which were cultured in high glucose DMEM complete medium (Biological Industries (BI)). The cells were cultured in a 5% $CO_2$ humidified incubator at 37 °C. New Zealand white rabbits (9–10 months old, 3.0–3.5 kg) and SD rats (2–3 months, 200–250 g) were obtained from the Laboratory Animal Center of Southern Medical University. The animals had access to food and water ad libitum and were hosted in ambient temperature 25 ± 1 °C, humidity

at 40–70%, dark/light cycle 6 p.m.–6 a.m. All the animal procedures were carried out under the guideline approved by the Institutional Animal Care and Use Committee (IACUC) of Southern Medical University (permit number: NFYY-2021-0847).

## Materials

Styrene (St, 99.5%), divinylbenzene (DVB, 80%), acrylic acid (AA, 99%), potassium perisulfite (APS, 98%), azodiethylbutyronitrile (AIBN, 99%), sodium dodecyl sulfate (SDS, 99%), 1-chlorodecane (CD, 99%), polyvinyl alcohol (PVA, $M_w = 20,000–30,000$, 88% hydrolysis) were purchased from Bailingwei Technology Co., Ltd. (Beijing, China). 3-(4,5-Dimethylthiazol-2-yl)–2,5-diphenyltetrazolium bromide (MTT, ≥98%) was acquired from Nanjing Keygen Biotech. Co., Ltd. (Nanjing, China). Lipiodol (containing iodine (I) 480 mg/mL) was purchased from Hengrui Pharmaceutical Co., Ltd. (Jiangsu, China). Dimethyl sulfoxide (DMSO), Dulbecco's modified eagle's medium (DMEM), PBS were purchased from Nanjing Keygen Biotech. Co., Ltd. (Nanjing, China). Cisplatin (Pt) was purchased from Shandong Boyuan Co., Ltd. (Shangdong, China). Oil red dye was purchased from Sigma-Aldrich. 8spheres® was purchased from Hengrui Pharmaceutical Co., Ltd. (China). The sizes of 8spheres® beads used in our study are 100–300 μm, similar with our Janus particle-engineered structural lipiodol droplets. In this experiment, we used deionized water (Millipore) with a resistivity of 18.2 MΩ·cm. Xylazine hydrochloride injection was purchased from Shengda Animal Pharmaceutical Co., Ltd. (Jilin, China).

## Preparation of Janus particles

The synthesis of Janus particles was as follows. First, 0.2 g of hydrophobic PS seeds with a particle size of 1.22 ± 0.12 μm were dispersed in 20.0 mL of SDS aqueous solution (0.25% W/V). Then the seed solution was incubated with 0.1 mL CD emulsion at 40 °C for 20 h. After that, 13 mL oil in water emulsion containing 1.5 mL of St, 1 mL of DVB, 0.5 mL of AA, and 40 mg of AIBN were added into the PS seed solution at 40 °C for 6 h. Subsequently, 5.0 mL of PVA aqueous solution (1% W/V) was added into the abovementioned solution and followed by deoxygenation bubbled with nitrogen for 15 min, which was finally polymerized at 70 °C for 14 h. After polymerization, the obtained mixture of Janus particles was centrifuged at 12,850 × $g$ for 10 min, and washed with ethanol and deionized water for three times, respectively. Finally, Janus particles were re-dispersed in deionized water and freeze-dried for subsequent use.

## Preparation of Cy7 modified Janus particles

Ten mg of EDC and 10 mg of NHS were added into 3 mL aqueous solution of Janus particles (7 mg/mL), respectively. And then, 10 μL of Cy7 dye was added into the abovementioned solution of Janus particles, which was stirred at 600 rpm for 24 h at room temperature. Finally, the obtained mixture of Janus particles was centrifuged at 12,850 × $g$ for 10 min, and washed with deionized water three times.

## Preparation of Janus particle-engineered structural lipiodol droplets

In a typical synthesis, 0.1 mL lipiodol was mixed with 6 mL aqueous solution of Janus particles (1.5 mg/mL), which was then sheared under 800 rpm at room temperature for 30 min. The obtained Janus particle-engineered structural lipiodol droplets were then cleaned with deionized water three times.

In our system, to facilitate observation, 50 μg of oil red was mixed into lipiodol, and other conditions were the same. The sizes of Janus particle-engineered structural lipiodol droplets could be controlled by tuning the concentration of Janus particles and the rate of lipiodol and deionized water.

## Characterization of Janus particle-engineered structural lipiodol droplets

The morphology of Janus particle-engineered structural lipiodol droplets was observed by an optical microscope and fluorescence microscope. Image J software was used to statistically analyze the particles size of Janus particle-engineered structural lipiodol droplets and their size distributions.

For CT images, the Janus particle-engineered structural lipiodol droplets with an average size of about 120 μm were used for CT scanning. The conditions were set as 120 KV, 350 mAs, window width 1588, window level 94, and unit HU. Three samples were placed in each group for scanning.

## Rheology of the Janus particle-engineered structural lipiodol droplets

All rheological measurements were performed with a strain-controlled MCR 302 rheometer (Anton Paar USA Inc., Torrance, CA). The viscosity values of Janus particle-engineered structural lipiodol droplets and lipiodol in the same volume (1 mL) were measured at 37 °C under different shear rates. Then the same volume (1 mL) of Janus particle-engineered structural lipiodol droplets and lipiodol were used to measure the storage modulus (G') values under different shear strains at 37 °C. When the aqueous solution of the Janus particle-engineered structural lipiodol droplets was added onto a stainless steel parallel plate of rheometer, the aqueous phase was removed, leaving the Janus particle-engineered structural lipiodol droplets for testing under different shear strains at 37 °C.

## Drug encapsulation of Janus particle-engineered structural lipiodol droplets

For drug encapsulation, 3 mg, 6 mg, 10 mg, and 20 mg of cisplatin were firstly mixed with 0.1 mL lipiodol, which was added into 6 mL aqueous solution of Janus particles (1.5 mg/mL) and subsequently was sheared under 800 rpm at room temperature for 30 min. The drug-loaded Janus particle-engineered structural lipiodol droplets were then cleaned with deionized water for three times. The loading efficiency of cisplatin in Janus particle-engineered structural lipiodol droplets was measured by an inductively coupled plasma optical emission spectrometer.

## Cytotoxicity assay in vitro

The cytotoxicity of Janus particle-engineered structural lipiodol droplets was determined by MTT experiment. HUVECs and HepG2 cells were incubated into a 96-well plate at a density of $5 \times 10^3$ cells/well, respectively. After 12 h culture, the Janus particle-engineered structural lipiodol droplets, Janus particles and lipiodol with different concentrations were added to each well of HUVECs and incubated for 48 h. For the cytotoxicity in HepG2 cells, the cisplatin-loaded Janus particle-engineered structural lipiodol droplets with different concentrations were added to each well of HepG2 cells and incubated for 48 h. Then, the medium was removed and replaced with fresh medium containing 110 μL MTT solution. After 4 h of incubation, the medium was removed and 150 μL DMSO was added into each well, which then was shaken for 5 min. The absorbance value was detected at 490 nm using a microplate reader (Synergy2, Bio-Tek, USA) to calculate the cell viability.

## The packing capacity of Janus particle-engineered structural lipiodol droplets and 8spheres® beads

The same number (about 500) of 8spheres® beads (the beads are dyed red for easy observation) and Janus particle-engineered structural lipiodol droplets were used for packing experiments. The Janus particle-engineered structural lipiodol droplets and 8spheres® beads were added in a semi-closed capillary (approximately inner diameter

was about 1 mm and 10 cm in length), respectively. After that, the packing height was calculated when the height did not change (about 6 h). The packing density of spheres is approximately 0.63[35].

### Hemolysis of Janus particle-engineered structural lipiodol droplets with different sizes

The anticoagulant whole blood of New Zealand white rabbits was collected via ear vein. Red blood cells were collected and incubated with various concentrations of Janus particle-engineered structural lipiodol droplets in tubes for 3 h at room temperature. And then, the samples were centrifuged at $955 \times g$ for 10 min and photographed. Finally, the absorbance of the supernatant at 541 nm was measured by using a microplate reader (Synergy2, Bio-Tek, USA).

### Injection test of Janus particle-engineered structural lipiodol droplets with different sizes

A mechanical tester (Instron, Norwood, MA) was used to test the injectable ability of the Janus particle-engineered structural lipiodol droplets. Briefly, we used Bluehill Version 4.25 software to record the force required for Janus particle-engineered structural lipiodol droplets (loaded into a 1 cc Medallion® syringe) passing through a 2.4 F, 80 cm catheter (Tokai Medical Products, Aichi, Japan) at a flow rate of 1 mL/min. The breaking force and injection force were analyzed.

### Embolization of Janus particle-engineered structural lipiodol droplets in a capillary glass tube

One mL of Janus particle-engineered structural lipiodol droplets were added into in a semi-closed capillary (inner diameter was about 1 mm) at a rate of 0.5 mL/min. The Janus particle-engineered structural lipiodol droplets were densely packed into the capillary tube. The embolization and viscoelastic deformation of Janus particle-engineered structural lipiodol droplets in capillary were observed under an optical microscope.

### Embolization experiment of Janus particle-engineered structural lipiodol droplets in a decellularized liver of SD rat

All animal testing procedures were reviewed and approved by the Animal Care and Use Committee of Southern Medical University (permit number: NFYY-2021-0847). All animal experiments were conducted in accordance with the regulations of China and Nanfang Hospital on the health and welfare of laboratory animals. The fresh liver was harvested by dissecting from the abdomen of the euthanized SD rats. And liver was perfused with distilled water (0.5 h), 4% TritonX-100 (3 h), and 0.5% sodium dodecyl sulfate (SDS) solution (12 h) in orders via the portal vein at a flow rate of 6 mL/min. Until the liver appeared transparent, 1 mL of the Janus particle-engineered structural lipiodol droplets were slowly injected into the decellularized liver via the portal vein at a flow rate of 40 μL/s to observe the embolization effectiveness in vitro. The viscoelastic deformation ability of the Janus particle-engineered structural lipiodol droplets was characterized by an optical microscope and CT images.

### Renal artery embolization of Janus particle-engineered structural lipiodol droplets in New Zealand white rabbits

All animal testing procedures were reviewed and approved by the Animal Care and Use Committee of Southern Medical University (permit number: NFYY-2021-0847). All animal experiments were conducted in accordance with the regulations of China and Nanfang hospitals on the health and welfare of laboratory animals. The experimental group was divided into three groups, with 3 male New Zealand rabbits in each group: Janus particle-engineered structural lipiodol droplets, 8spheres® microspheres, and Lipiodol-based emulsion. A water-in-oil emulsion of lipiodol-based emulsion (O/W ratio is 2:1 v/v) was prepared by emulsification. Digitally subtracted angiography (DSA) was used to guide all interventional surgeries under aseptic conditions. New Zealand rabbits were fed under standard conditions in animal houses. After fasting for 12 h, the rabbits were injected intramuscularly with xylazine hydrochloride (0.2 mL/kg) for anesthesia. After general anesthesia, the rabbits were fixed on the worktops in a supine position. Then, a guidewire and catheter sheath was introduced into the femoral artery in the right abdominal femoral of rabbits, followed by introducing a microguidewire and microcatheter into the catheter sheath. After that, 3 mL of iohexol was injected with a syringe at a rate of 1.0 mL/s to observe the distribution of the abdominal vessels. Subsequently, the microcatheter was slowly pushed into the renal artery and following angiography was performed to carefully confirm the blood vessels of the right kidney once again. Finally, each rabbit was injected with the same volume of 8spheres®, Lipiodol-based emulsion, and Janus particle-engineered structural lipiodol droplets at a rate of 0.25 mL/s. After injection of 30 min, the blood vessels of rabbits was performed for DSA images to check the recanalization of blood vessels on day 0 post-embolization.

After 14 days post-embolization, 3 mL of contrast agent (iohexol) was injected into blood vessels for DSA evaluation, and the embolization effectiveness was examined by CT arteriography. Moreover, after 14 days post-embolization, other organs, such as heart, liver, spleen, lung, left kidney, and brain were also examined by CT. Blood samples were collected for biochemical tests and pro-inflammatory cytokines (IL-1β, IL-6, PCT, and TNF-α) tests. And then the New Zealand rabbits were euthanized. Finally, the right kidney and left kidney of each group were harvested to further evaluate the embolization effectiveness.

### Pathological assessment of the embolization effectiveness of Janus particle-engineered structural lipiodol droplets

After dissecting the left and right kidney of rabbit, half of the kidney samples were immediately placed in sterile tubes and stored in an −80 °C refrigerator for subsequent frozen sections. The other half was placed in a 4% paraformaldehyde solution and then was performed by paraffin sections. The frozen sections and paraffin sections were cut into 4 μm thick. H&E staining was performed using standard clinical laboratory protocol. All sections were observed under an optical microscope.

### Statistical analysis

Statistical analysis was performed with PRISM 8 (GraphPad Software, San Diego, CA). All results were presented as mean ± SD. One-way ANOVA or unpaired two-tailed Student's $t$ tests was used to establish statistical significance using IBM SPSS 19.0 software.

### Reporting summary

Further information on research design is available in the Nature Portfolio Reporting Summary linked to this article.

## Data availability

The source data generated in this study are provided in the Supplementary Information/Source data file. All other data are available from the corresponding author upon request. Source data are provided with this paper.

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

## Acknowledgements

This work was financially supported by the grants from the National Natural Science Foundation of China (22075127, 22275080) and Natural Science Foundation of Guangdong Province (2022A1515012044, 2023A1515011131).

## Author contributions

J.-B.F., W.Z., and J.H. conceived and designed the experiments. S.T. performed most of the experiments and data analysis. J.-B.F. and S.T. analyzed the data and wrote the manuscript. B.L. and H.Z. assisted with the in vitro experiments. J.-B.F., W.Z., X.H., S.T., and J.Z. assisted with the in vivo animal experiments. All authors discussed the results and revised and commented on the manuscript.

## Competing interests

The authors declare no competing interests.
