## [Peer Review File · Nature Communications]

Reviewers' Comments:

Reviewer #1:

Remarks to the Author:

The manuscript under consideration describes the construction of a new embolic material for arterial embolization. This is a very original study where for the first time lipiodol droplets have been formulated in a stable manner, without leakage of lipiodol and capable of being packaged in the form of particles for a potential clinical application. The approach used by the authors is fundamentally different from those usually found where one tries to stabilize a lipiodol emulsion rather than to formulate stable lipiodol droplets. However, although this embolic material is remarkable for its innovation, the question arises of its relevance with the real needs of interventional radiologists. Besides this main comment, the lack of information on the methodology and the imprecision of some data sometimes lead to comprehension difficulties that hinders the quality of this study.

If the authors can address the concerns below the paper should be published in Nature Comm.

Major concerns that needs to be address prior to publication:

1- Today, the current demand is not especially towards a permanent embolization or an improvement of the mechanical properties as claimed by the authors in this manuscript (see doi.org/10.1186/s12951-023-01804-7 and doi.org/10.1016/j.actbio.2022.07.019). Indeed, permanent occlusion may prevent retreatment, and post-embolization syndrome should be considered. In addition, the deformation of the "Janus particle-engineered lipiodol droplets" results in more distal penetration than expected after their injection, precluding the ability to occlude vessels precisely with the use of a proper particle size range. Is this new material really suitable for the current needs in embolization? A clear rationale must be introduced to show the interest of this material which at first sight is far from obvious. It would be appropriate to focus the manuscript only on chemoembolization and not embolization in general where particle size and occlusion level is important. Moreover, the authors are invited to moderate their remarks on the efficiency of the current products throughout the manuscript.

2- Janus particles are produced from styrene and divinyl benzene. What will be the long-term fate of these non-degradable particles in the body?

3- Cisplatin was successfully encapsulated in the oily core of the Janus particle-engineered lipiodol droplets. Which amount of Janus particle-engineered lipiodol droplets should be injected to achieve a therapeutic dose of cisplatin (considered today at 100-140 mg per injection)? A release profile should be added?

Minor concerns:

4- Cytotoxicity was studied up to 32 µg/ml. After how long of time of incubation were these measurements carried out? Does this concentration reflect the quantities that will be injected in humans for embolization?

5- As shown by the authors, 10 g of Janus particle-engineered lipiodol droplets can be formulated per batch. On average, 2 ml of microspheres are injected per embolization (or 10-12 ml of lipiodol emulsion), can the envisaged process really be considered to produce enough material at large scale?

6- Which size of Janus particle-engineered lipiodol droplets were injected on rabbits? Which Lipiodol emulsion was injected (W/O ratio, type of emulsion)? How much of each product (emulsion, microspheres and Janus particle-engineered lipiodol droplets) was injected into the rabbit? After how long the post-embolization control was performed at D0? It is well-known that microspheres after embolization evolve in the vascular network and that re-opacification of the proximal branches appears after 20-30 min. This does not mean that there is recanalization.

7- Injectability tests should have been performed with the various sizes of Janus particle-engineered lipiodol droplets.

8- How was injected the Janus particle-engineered lipiodol droplets in the decellularized liver?

Reviewer #2:

Remarks to the Author:

In the study 'Janus Particle-Engineered Structural Lipiodol Droplets for Arterial Embolization', the authors present a new embolic agent based on Lipiodol self-assembled Janus droplets. These viscoelastic particles have excellent deformability and imageability. They also have drug carrying

capability. The authors tested embolic efficacy both in vitro using decellularized mouse liver and in vivo using rabbit renal models. The idea is novel and the overall study is sound. The reviewer has the following comments.

1. The size distribution for these particles (i.e., 120 ± 60) is huge, as compared to clinically used microbeads. Companies strive for uniform size distribution. The inhomogeneity is rather undesired as it can lead to difficulties in particle selection, administration with lack of prediction for embolic efficacy. How can this issue be addressed?
2. The particles appear to be broken as shown in the SEM image of SI Fig. 1. How stable and mechanically intact these particles are?
3. SI Fig. 18 and 19, the big gaps noticed on H&E images may be due to histology processing, which can influence the actual particle distribution (i.e., loss of particles during the process). The disintegration between the particles and tissue may be due to various factors which should be more rigorously discussed. Therefore, it may not be conclusive to compare the occlusion efficacy based on histology images.
4. Are the droplets biodegradable? What's their fate in vivo in short and long term?
5. The authors suggest that the 8spheres may not be as viscoelastic as compared to the newly fabricated droplets. Mechanical testing needs to be carried out for 8spheres to support the argument for comparison.
6. Have the authors look into additional tissue response besides H&E? Any inflammation or fibrosis in the microenvironment?
7. The supplementary video description (in SI) does not match the video content. There is no in vitro retrieval. For the two videos, both appear to have materials leaking out, causing distal dislodgement that can be dangerous.
8. There is no hemocompatibility testing. The interaction of any newly developed embolic material and blood is critical to investigate.

Reviewer #3:

Remarks to the Author:

The authors presented the Janus Particle-Engineered Structural Lipidol Droplets for Arterial Embolization. In the first part of the paper, the authors employed their previous synthetic technique to prepare the Janus particles.

Further, they used the prepared Janus particles for arterial embolization (liver and kidneys). The embolization results are quite promising as some other studies, but the synthesis mechanism and justification (structure-property relationship) show limited novelty.

The self-assembly of macro/microparticles is easy and has much literature available, but the self-assembly of Janus NPs at the nanoscale is very challenging.

Moreover, characterization of Janus is also limited in this study, for example, SEM did not show any Janus structure, but it's quite easy to obtain SEM of Janus microparticles, also, the degree of Janus formation should be explained clearly.

In addition, the structural properties are not consistent in the capillary glass tube, and the uniformity of the structure is limited at a large scale.

The ratio of cisplatin encapsulation needed to be explained (No of Particles per concentration of cisplatin). The above mention concerns showed limited novelty to be published in this journal.

List of Responses to the Reviews

Thanks a lot for the reviewers' comments.

Response to Reviewer 1:

The manuscript under consideration describes the construction of a new embolic material for arterial embolization. This is a very original study where for the first time lipiodol droplets have been formulated in a stable manner, without leakage of lipiodol and capable of being packaged in the form of particles for a potential clinical application. The approach used by the authors is fundamentally different from those usually found where one tries to stabilize a lipiodol emulsion rather than to formulate stable lipiodol droplets. However, although this embolic material is remarkable for its innovation, the question arises of its relevance with the real needs of interventional radiologists. Besides this main comment, the lack of information on the methodology and the imprecision of some data sometimes lead to comprehension difficulties that hinders the quality of this study.

If the authors can address the concerns below the paper should be published in Nature Comm.

Response: Thanks very much for the reviewer's highly positive comments and excellent suggestions. We have discussed the relevance of the Janus particle-engineered structural lipiodol droplets with the real needs of interventional radiologists. According to the reviewer's kind suggestion, we have focused the manuscript on chemoembolization. And also, we have supplemented the information on the methodology and revised the imprecision of some data.

Major concerns that needs to be address prior to publication:

Comment 1: Today, the current demand is not especially towards a permanent embolization or an improvement of the mechanical properties as claimed by the authors in this manuscript (see doi.org/10.1186/s12951-023-01804-7 and doi.org/10.1016/j.actbio.2022.07.019). **(1)** Indeed, permanent occlusion may prevent retreatment, and post-embolization syndrome should be considered. **(2)** In addition, the deformation of the "Janus particle-engineered structural lipiodol droplets" results in

more distal penetration than expected after their injection, precluding the ability to occlude vessels precisely with the use of a proper particle size range. Is this new material really suitable for the current needs in embolization? A clear rationale must be introduced to show the interest of this material which at first sight is far from obvious. It would be appropriate to focus the manuscript only on chemoembolization and not embolization in general where particle size and occlusion level is important. (3) Moreover, the authors are invited to moderate their remarks on the efficiency of the current products throughout the manuscript.

Response: Thanks very much for the reviewers' good comments.

(1) We agree with the reviewer's viewpoints that permanent occlusion may prevent retreatment and cause syndrome. Like liquid lipiodol, the Janus particle-engineered structural lipiodol droplets are not permanent occlusion. We have investigated the short-term and long-term fate of the Janus particle-engineered structural lipiodol droplets post-embolization. The results indicated that accompanying with the volume reduction of kidney post-embolization, the Janus particle-engineered structural lipiodol droplets could be gradually metabolized, regardless of lipiodol or Janus particles. We monitored the different organs by CT images on day 0, day 14, day 30, and day 45 post-embolization to further explore the metabolic pathway of lipiodol within the Janus particle-engineered structural lipiodol droplets. The results demonstrated that except for kidney and bladder, no evidence of lipiodol distribution was observed in the liver, heart, spleen, lung, brain, suggesting that lipiodol was mainly metabolized *via* kidney. We also investigated the metabolic pathway of Janus particles within the Janus particle-engineered structural lipiodol droplets. To monitor the fate of these non-degradable Janus particles, they were chemically modified with Cy7 fluorescent dyes. The results indicated that except for kidney, Janus particles within the Janus particle-engineered structural lipiodol droplets could also be observed in liver and bladder on 45 days post-embolization. And, in the collected urine of rabbits, we also clearly observed the Janus particles. These results indicated the metabolization of Janus particles mainly proceeded with liver and kidney.

We have added the data and detailed discussions of the short-term and long-term fate

of Janus particle-engineered structural lipiodol droplets in the revised manuscript.

In Page 13, Line 322-332; Page 14, Line 333-348

“Next, we investigated the short-term and long-term fate of the Janus particle-engineered structural lipiodol droplets post-embolization. The results indicated that accompanying with the volume reduction of kidney, the Janus particle-engineered structural lipiodol droplets could be gradually metabolized, regardless of Janus particles or lipiodol. As shown in Supplementary Fig. 26, the lipiodol intensity was gradually decreased with the prolonging of embolization time, indicating that the lipiodol within the Janus particle-engineered structural lipiodol droplets could be slowly metabolized. To further explore the metabolic pathway of lipiodol within the Janus particle-engineered structural lipiodol droplets, we monitored the different organs by CT images on day 0, day 14, day 30, and day 45 post-embolization. The results demonstrated that except for kidney and bladder, no evidence of lipiodol distribution was observed in the lung, liver, heart, spleen, brain, and normal left kidney, suggesting that lipiodol was mainly metabolized *via* kidney (Supplementary Fig. 27).

We also investigated the metabolic pathway of Janus particles within the Janus particle-engineered structural lipiodol droplets. To monitor the fate of these non-degradable Janus particles, they were chemically modified with Cy7 fluorescent dyes. And then, these fluorescent Janus particles were used to fabricate Janus particle-engineered structural lipiodol droplets. To assess the biodistributions of Janus particles within the Janus particle-engineered structural lipiodol droplets, rabbits were sacrificed on day 45 post-embolization and the main organs were carefully excised for fluorescence imaging using an *in vivo* imaging system (IVIS Lumina II, Caliper, USA). The results indicated that except for kidney, the fluorescence of Janus particles within the Janus particle-engineered structural lipiodol droplets was also observed in liver and bladder on 45 days post-embolization (Supplementary Fig. 28). And, in the collected urine of rabbits, we also clearly observed the Janus particles (Supplementary Fig. 29). These results indicated the metabolic pathway of Janus particles mainly proceeded with liver and kidney.”

Supplementary Fig 26. The lipiodol intensity within Janus particle-engineered structural lipiodol droplets in rabbits at different times post-embolization. Accompanying with the volume reduction of kidney, lipiodol within the Janus particle-engineered structural lipiodol droplets could be metabolized gradually.

Supplementary Fig 27. CT images of the main organs at different times post-embolization. The results demonstrated that except for right kidney and bladder, no evidence of lipiodol was observed in the lung, liver, heart, spleen, brain, and normal left kidney, suggesting that lipiodol was mainly metabolized *via* kidney.

Supplementary Fig 28. The biodistributions of Janus particle-engineered structural lipiodol droplets on day 45 post-embolization. Rabbits were sacrificed on day 45 post-embolization and the main organs were carefully excised to observe the biodistributions of Janus particles. The result indicated that except for kidney, the fluorescence of Janus particles within the Janus particle-engineered structural lipiodol droplets was also observed in liver and bladder. These results indicated the metabolization of Janus particles mainly proceeded with liver and kidney.

Supplementary Fig 29. TEM images of the Janus particles in the collected urine of rabbit post-embolization. In the collected urine of rabbit, we also clearly observed the Janus particles.

(2) We appropriate so much for reviewer’s kind suggestion. According to the suggestion, we have focused the manuscript on chemoembolization. Please see the detailed information in the revised manuscript.

In Page 2, Line 41-52; Page 3, Line 53-58

“A variety of embolic materials (represented by solid beads and liquid embolic materials) have been widely used in clinical practice^{5, 7-9}, especially in tumor chemoembolization. However, owing to the different sizes (from 5-10 μm to 1-2 cm in

diameter) and complex architectures of blood vessels, most embolic materials remain limited by efficient chemoembolization^{5,10}. In the arterial chemoembolization of tumors, such as transarterial chemoembolization (TACE), beyond embolization of large-sized arterial vasculatures, the embolic materials are also expected to travel distally to embolize those finer arterial vasculatures to achieve distal embolization, which is beneficial to enhance the efficiency of chemoembolization¹¹⁻¹³. However, the solid beads with sizes more than 100 μm are usually limited by embolizing the finer arterial vasculatures probably because of their poor viscoelastic deformation abilities¹⁴⁻¹⁷. Liquid embolic materials, such as lipiodol, are promising to flow to finer vasculatures and show a good capacity for radiography and drug loading, which has attracted particular attention for tumor chemoembolization over the past several decades^{18,19}. However, these lipiodol systems are highly unstable in the blood and remain limited by, e.g., their recanalization, non-specific embolization, toxicity, etc²⁰⁻²³. Therefore, there is a critical need for new embolic materials that can adapt to the architectures of the blood vessels for efficient arterial embolization.”

(3) According to the suggestion, we have moderated the remarks on the efficiency of the current products throughout the manuscript.

Comment 2: Janus particles are produced from styrene and divinyl benzene. What will be the long-term fate of these non-degradable particles in the body?

Response: Thanks very much for the reviewers’ good comments. We have investigated the long-term fate of Janus particles within the Janus particle-engineered structural lipiodol droplets in rabbits post-embolization. The metabolic pathway of Janus particles within the Janus particle-engineered structural lipiodol droplets was investigated. To monitor the fate of these non-degradable Janus particles, they were chemically modified with Cy7 fluorescent dyes. The results indicated that except for kidney, the Janus particles within the Janus particle-engineered structural lipiodol droplets could also be observed in liver and bladder on 45 days post-embolization. And, in the collected urine of rabbits, we also clearly observed the Janus particles. These results indicated the metabolization of Janus particles mainly proceeded with liver and kidney.

We have added the data and detailed discussions of the long-term fate of Janus particles within Janus particle-engineered structural lipiodol droplets in the revised manuscript.

In Page 14, Line 336-348

“We also investigated the metabolic pathway of Janus particles within the Janus particle-engineered structural lipiodol droplets. To monitor the fate of these non-degradable Janus particles, they were chemically modified with Cy7 fluorescent dyes. And then, these fluorescent Janus particles were used to fabricate Janus particle-engineered structural lipiodol droplets. To assess the biodistributions of Janus particles within the Janus particle-engineered structural lipiodol droplets, rabbits were sacrificed on day 45 post-embolization and the main organs were carefully excised for fluorescence imaging using an in vivo imaging system (IVIS Lumina II, Caliper, USA). The results indicated that except for kidney, the fluorescence of Janus particles within the Janus particle-engineered structural lipiodol droplets was also observed in liver and bladder on 45 days post-embolization (Supplementary Fig. 28). And, in the collected urine of rabbits, we also clearly observed the Janus particles (Supplementary Fig. 29). These results indicated the metabolic pathway of Janus particles mainly proceeded with liver and kidney.”

Supplementary Fig 28. The biodistributions of Janus particle-engineered structural lipiodol droplets on day 45 post-embolization. Rabbits were sacrificed on day 45 post-embolization and the main organs were carefully excised to observe the biodistributions of Janus particles. The result indicated that except for kidney, the fluorescence of Janus particles within the Janus particle-engineered structural lipiodol droplets was also observed in liver and bladder on day 45 post-embolization. These results indicated the metabolization of Janus particles mainly proceeded with liver and kidney.

Supplementary Fig 29. TEM images of the Janus particles in the collected urine of rabbit post-embolization. In the collected urine of rabbits, we also clearly observed the Janus particles.

Comment 3: Cisplatin was successfully encapsulated in the oily core of the Janus particle-engineered structural lipiodol droplets. Which amount of Janus particle-engineered structural lipiodol droplets should be injected to achieve a therapeutic dose of cisplatin (considered today at 100-140 mg per injection)? A release profile should be added?

Response: Thanks very much for the reviewer's comments. We have calculated that 1 mL lipiodol can produce 6000 number of Janus particle-engineered structural lipiodol droplets with size of $120 \pm 40 \mu\text{m}$. The encapsulation of cisplatin within the Janus particle-engineered structural lipiodol droplets could be controlled by tuning the feed amount of cisplatin. For example, in a typical encapsulation, the encapsulation efficiency of cisplatin within Janus particle-engineered structural lipiodol droplets (containing 0.1 mL of lipiodol, approximately 600 droplets) was approximately 93% at the feed amount of 6 mg. According to the therapeutic dose of cisplatin (100-140 mg per injection), approximately 10000-15000 number of Janus particle-engineered structural lipiodol droplets (containing approximately 1.8 mL-2.5 mL of lipiodol) should be injected.

According to the reviewer's suggestion, we performed the release profile of Janus

particle-engineered structural lipiodol droplets. The results demonstrated that the obtained cisplatin-loaded Janus particle-engineered structural lipiodol droplets exhibited pH-responsive drug release due to the carboxyl group of Janus particles. We have added the release profile of Janus particle-engineered structural lipiodol droplets in the revised manuscript, as shown in Supplementary Fig 7.

In Page 6, Line 137-141

“We performed the release profile of the Janus particle-engineered structural lipiodol droplets. The results demonstrated that the obtained cisplatin-loaded Janus particle-engineered structural lipiodol droplets exhibited pH-responsive drug release due to the carboxyl group of Janus particles (Supplementary Fig. 7)”.

Supplementary Fig 7. The cumulative release of cisplatin within Janus particle-engineered structural lipiodol droplets.

Minor concerns:

Comment 4: Cytotoxicity was studied up to 32 $\mu\text{g/ml}$. After how long of time of incubation were these measurements carried out? Does this concentration reflect the quantities that will be injected in humans for embolization?

Response: Thanks very much for the reviewer’s comments. The cytotoxicity was measured after 48 h incubation. We have added the detailed information in the revised Supplementary Information.

In Page 3, Line 64-65; Page 4, Line 66-77 in Supplementary Information

“The cytotoxicity of Janus particle-engineered structural lipiodol droplets was

determined by MTT experiment. HUVEC cells and HepG2 cells were incubated into a 96-well plate at a density of 5×10^3 cells/well, respectively. After 12 h culture, the Janus particle-engineered structural lipiodol droplets, Janus particles and lipiodol with different concentrations were added to each well of HUVEC cells and incubated for 48 h. For the cytotoxicity of Janus particle-engineered structural lipiodol droplets HepG2 cells, the Janus particle-engineered structural lipiodol droplets with different concentrations were added to each well of HepG2 cells and incubated for 48 h. Then, the medium was removed and replaced with fresh medium containing 110 μ L MTT solution. After 4 h of incubation, the medium was removed and 150 μ L DMSO was added into each well, which then was shaken for 5 minutes. The absorbance value was detected at 490 nm using a microplate reader (Synergy2, Bio-Tek, USA) to calculate the cell viability.”

We compared the cytotoxicity of cisplatin and cisplatin-loaded Janus particle-engineered structural lipiodol droplets. The results indicated the cytotoxicity of cisplatin-loaded Janus particle-engineered structural lipiodol droplets was lower than the cisplatin at the concentration down to 16 μ g/mL, while it was comparable to the cisplatin at 32 μ g/mL. The concentration is only at cellular level, which needs more evidence to verify its effectiveness, such as first on animal models and next to humans. It is difficult for us in current stage to conclude that the concentration reflects the quantities that will be injected into humans for embolization.

The cytotoxicity comparison of the cisplatin and cisplatin-loaded Janus particle-engineered structural lipiodol droplets.

Comment 5: As shown by the authors, 10 g of Janus particle-engineered structural lipiodol droplets can be formulated per batch. On average, 2 ml of microspheres are injected per embolization (or 10-12 ml of lipiodol emulsion), can the envisaged process really be considered to produce enough material at large scale?

Response: Thanks very much for the reviewer's comments. We attempted to use 20 mL of lipiodol to produce Janus particle-engineered structural lipiodol droplets. As shown in Supplementary Fig 9, approximately 25g of Janus particle-engineered structural lipiodol droplets could be produced per batch. We have updated the results in the revised manuscript, as shown in Supplementary Fig 9.

In Page 6, Line 144-147

“When the fabrication was scaled up to 200 times that of the aforementioned feed, this approach produced approximately 25 g of Janus particle-engineered structural lipiodol droplets in one batch, providing an effective way to produce the structural lipiodol droplets at a large scale”.

Supplementary Fig 9. Large-scale fabrication of Janus particle-engineered structural lipiodol droplets. The lipiodol was stained with oil red.

Comment 6: Which size of Janus particle-engineered structural lipiodol droplets were injected on rabbits? Which Lipiodol emulsion was injected (W/O ratio, type of emulsion)? How much of each product (emulsion, microspheres and Janus particle-engineered structural lipiodol droplets) was injected into the rabbit? After how long the

post-embolization control was performed at D0? It is well-known that microspheres after embolization evolve in the vascular network and that re-opacification of the proximal branches appears after 20-30 min. This does not mean that there is recanalization.

Response: Thanks very much for the reviewer's comments. In the study, the Janus particle-engineered structural lipiodol droplets with sizes of $120 \pm 40 \mu\text{m}$ were used to perform the embolization in rabbits. A water-in-oil of lipiodol emulsion (O/W ratio is 2:1 v/v) was injected. 1.5 mL of 8spheres® beads (approximately 30000 beads), lipiodol-based emulsion (containing 1 mL lipiodol), and Janus particle-engineered structural lipiodol droplets (containing 1 mL lipiodol, approximately 6000 droplets) were injected into the rabbits, respectively.

We have added the information in the revised manuscript.

In Page 5, Line 121-123

“We used the Janus particle-engineered structural lipiodol droplets with sizes of $120 \pm 40 \mu\text{m}$ to perform all the subsequent experiments.”

In Page 19, Line 467-468

“A water-in-oil emulsion of lipiodol-based emulsion (O/W ratio is 2:1 v/v) was prepared by emulsification.”

In Page 9, Line 229-230; Page 10, Line 231-233

“1.5 mL of embolic materials, including Janus particle-engineered structural lipiodol droplets (containing 1 mL lipiodol, approximately 6000 droplets), clinical 8spheres®beads (approximately 30000 beads), and lipiodol-based emulsion (containing 1 mL lipiodol), were injected into the right kidney of rabbits via the 2.2F coaxial catheter, respectively”.

At day 0, the post-embolization control was performed for DSA images after injection of 30 min. As shown in Fig. 4a, at day 0 post-embolization, the DSA results indicated that the blood vessels could hardly be observed in the cases administered with Janus particle-engineered structural lipiodol droplets and clinical 8spheres® beads, suggesting their good embolization capacities.

We have added the information in the revised manuscript.

In Page 19, Line 480-481

“After injection of 30 min, the blood vessels of rabbits was performed for DSA images to check the recanalization of blood vessels on day 0 post-embolization.”

Comment 7: Injectability tests should have been performed with the various sizes of Janus particle-engineered structural lipiodol droplets.

Response: Thanks very much for the reviewer’s constructive comments. According to the reviewer’s suggestion, we performed injectability tests of Janus particle-engineered structural lipiodol droplets with different sizes. The result demonstrated that the breakloose force and injection force of all the Janus particle-engineered structural lipiodol droplets with different sizes was less than 10 N, suggesting good transcatheter injection. We have supplemented injectability tests and discussion in the revised manuscript, as shown in Fig. 3b.

In Page 7, Line 172-175

“Moreover, we also performed injectability tests of Janus particle-engineered structural lipiodol droplets with different sizes. The result demonstrated that the breakloose force and injection force of all the Janus particle-engineered structural lipiodol droplets was less than 10 N, suggesting good transcatheter injection (Fig. 3b)”

Fig. 3b. The injection force curves of Janus particle-engineered structural lipiodol droplets with different sizes.

Comment 8: How was injected the Janus particle-engineered structural lipiodol

droplets in the decellularized liver?

Response: Thanks very much for the reviewer's comments. The Janus particle-engineered structural lipiodol droplets were injected into the decellularized liver *via* portal vein at a flow rate of 40 $\mu\text{L/s}$. We have added the information in the revised manuscript.

In Page 18, Line 453-456

“Until the liver appeared transparent, 1 mL of the Janus particle-engineered structural lipiodol droplets were slowly injected into the decellularized liver *via* the portal vein at a flow rate of 40 $\mu\text{L/s}$ to observe the embolization effectiveness *in vitro*.”

Response to Reviewer 2:

In the study 'Janus Particle-Engineered Structural Lipiodol Droplets for Arterial Embolization', the authors present a new embolic agent based on Lipiodol self-assembled Janus droplets. These viscoelastic particles have excellent deformability and imageability. They also have drug carrying capability. The authors tested embolic efficacy both *in vitro* using decellularized mouse liver and *in vivo* using rabbit renal models. The idea is novel and the overall study is sound. The reviewer has the following comments.

Response: Thanks very much for the reviewer's highly positive comments and excellent suggestions.

Comment 1: The size distribution for these particles (i.e., 120 ± 60) is huge, as compared to clinically used microbeads. Companies strives for uniform size distribution. The inhomogeneity is rather undesired as it can lead to difficulties in particle selection, administration with lack of prediction for embolic efficacy. How can this issue be addressed?

Response: Thanks very much for the reviewer's good comments. We agree with the reviewer's viewpoints that uniform size distribution of embolic agent is desired. To address this issue, the microfluidic technique may be a promising strategy. We have attempted to produce uniform Janus particle-engineered structural lipiodol droplets by microfluidic technique. In a typical fabrication, when a lipiodol droplet was extruded

from a dispersion phase channel, the Janus particles in a continuous phase channel can self-assemble onto the interface of lipiodol droplets to fabricate Janus particle-engineered structural lipiodol droplets with uniform size distribution. We have added the data and made a discussion in the Discussion Section in the revised manuscript.

In Page 14, Line 350-358; Page 15, Line 359-361

“Since 1978, Prof. Yamada proposed the transcatheter arterial chemoembolization technique, and lipiodol-based embolization has been widely used in the clinic. In our study, lipiodol was formulated into droplets in a stable manner by programming the self-assembly of Janus particles at the lipiodol-water interface. In our current study, although efficient embolization was achieved by these Janus particle-engineered structural lipiodol droplets, their wide size distribution ($120 \pm 40 \mu\text{m}$) may lead to difficulties in particle selection and administration with a lack of prediction for embolic efficacy. To address this issue, we attempted to produce uniform Janus particle-engineered structural lipiodol droplets by microfluidic technique. In a typical fabrication, when a lipiodol droplet was extruded from a dispersion phase channel, the Janus particles in a continuous phase channel can self-assemble onto the interface of lipiodol droplets to fabricate Janus particle-engineered structural lipiodol droplets with uniform size distribution ($127 \pm 7 \mu\text{m}$) (Supplementary Fig 30).”

Supplementary Fig 30. Fabrication of uniform Janus particle-engineered structural lipiodol droplets by microfluidic technique.

Comment 2: The particles appear to be broken as shown in the SEM image of SI Fig. 1. How stable and mechanically intact these particles are?

Response: Thanks very much for the reviewer's comments. Supplementary Fig 1a is the morphology of Janus particles. The Janus particles exhibited hemispherical shapes, into which the convex surface of the Janus particle was hydrophilic poly(acrylic acid) and the concave surface was hydrophobic poly(styrene-co-divinyl benzene). These Janus particles have good mechanical stability because they possessed stable structure networks by chemically crosslinking polymerization during the process of synthesis, as indicated by our previous study (*Sci. Adv.* 2017, 3, e1603203; *Macromolecules* 2019, 52, 3237).

Comment 3: SI Fig. 18 and 19, the big gaps noticed on H&E images may be due to histology processing, which can influence the actual particle distribution (i.e., loss of particles during the process). The disintegration between the particles and tissue may be due to various factors which should be more rigorously discussed. Therefore, it may not be conclusive to compare the occlusion efficacy based on histology images.

Response: Thanks very much for the reviewer's good comments. We agree with the reviewer's viewpoints that the histology processing may influence the actual particle distribution. In our study, the embolization effectiveness was evaluated by pathological analysis in Fig. 5a and CT images in Fig. 4. To avoid misunderstanding, we have revised the descriptions in the revised manuscript.

In Page 13, Line 316-321

“In contrast, the 8spheres® beads were mainly observed in those large-sized arterial vasculatures and we did not observe them in the finer vasculatures (Supplementary Fig. 24). In the lipiodol-based emulsion group, the embolized right kidney exhibited poor efficiency after 14 days post-embolization and most regions remained maintained normal tissue structures (Fig. 5a and Supplementary Fig. 25)”

Comment 4: Are the droplets biodegradable? What's their fate in vivo in short and long

term?

Response: Thanks very much for the reviewer's good comments. We have investigated the short-term and long-term fate of the Janus particle-engineered structural lipiodol droplets post-embolization. The results indicated that accompanying with the volume reduction of kidney post-embolization, the Janus particle-engineered structural lipiodol droplets could be gradually metabolized, regardless of lipiodol or Janus particles. We monitored the different organs by CT images on day 0, day 14, day 30, and day 45 post-embolization to further explore the metabolic pathway of lipiodol within the Janus particle-engineered structural lipiodol droplets. The results demonstrated that except for kidney and bladder, no evidence of lipiodol distribution was observed in the liver, heart, spleen, lung, brain, suggesting that lipiodol was mainly metabolized *via* kidney. We also investigated the metabolic pathway of Janus particles within the Janus particle-engineered structural lipiodol droplets. To monitor the fate of these non-degradable Janus particles, they were chemically modified with Cy7 fluorescent dyes. The results indicated that except for kidney, Janus particles within the Janus particle-engineered structural lipiodol droplets could also be observed in liver and bladder on 45 days post-embolization. And, in the collected urine of rabbits, we also clearly observed the Janus particles. These results indicated the metabolization of Janus particles mainly proceeded with liver and kidney.

We have added the data and detailed discussions of the short-term and long-term fate of Janus particle-engineered structural lipiodol droplets in the revised manuscript.

In Page 13, Line 322-332; Page 14, Line 333-348

“Next, we investigated the short-term and long-term fate of the Janus particle-engineered structural lipiodol droplets post-embolization. The results indicated that accompanying with the volume reduction of kidney, the Janus particle-engineered structural lipiodol droplets could be gradually metabolized, regardless of Janus particles or lipiodol. As shown in Supplementary Fig. 26, the lipiodol intensity was gradually decreased with the prolonging of embolization time, indicating that the lipiodol within the Janus particle-engineered structural lipiodol droplets could be slowly metabolized. To further explore the metabolic pathway of lipiodol within the Janus particle-

engineered structural lipiodol droplets, we monitored the different organs by CT images on day 0, day 14, day 30, and day 45 post-embolization. The results demonstrated that except for kidney and bladder, no evidence of lipiodol distribution was observed in the lung, liver, heart, spleen, brain, and normal left kidney, suggesting that lipiodol was mainly metabolized *via* kidney (Supplementary Fig. 27).

We also investigated the metabolic pathway of Janus particles within the Janus particle-engineered structural lipiodol droplets. To monitor the fate of these non-degradable Janus particles, they were chemically modified with Cy7 fluorescent dyes. And then, these fluorescent Janus particles were used to fabricate Janus particle-engineered structural lipiodol droplets. To assess the biodistributions of Janus particles within the Janus particle-engineered structural lipiodol droplets, rabbits were sacrificed on day 45 post-embolization and the main organs were carefully excised for fluorescence imaging using an *in vivo* imaging system (IVIS Lumina II, Caliper, USA). The results indicated that except for kidney, the fluorescence of Janus particles within the Janus particle-engineered structural lipiodol droplets was also observed in liver and bladder on 45 days post-embolization (Supplementary Fig. 28). And, in the collected urine of rabbits, we also clearly observed the Janus particles (Supplementary Fig. 29). These results indicated the metabolic pathway of Janus particles mainly proceeded with liver and kidney.”

Supplementary Fig 26. The lipiodol intensity within Janus particle-engineered structural lipiodol droplets in rabbits at different times post-embolization. Accompanying with the volume reduction of kidney, lipiodol within the Janus particle-

engineered structural lipiodol droplets could be metabolized gradually.

Supplementary Fig 27. CT images of the main organs at different times post-embolization. The results demonstrated that except for right kidney and bladder, no evidence of lipiodol was observed in the lung, liver, heart, spleen, brain, and normal left kidney, suggesting that lipiodol was mainly metabolized *via* kidney.

Supplementary Fig 28. The biodistributions of Janus particle-engineered structural lipiodol droplets on day 45 post-embolization. Rabbits were sacrificed on day 45 post-embolization and the main organs were carefully excised to observe the biodistributions of Janus particles. The result indicated that except for kidney, the fluorescence of Janus particles within the Janus particle-engineered structural lipiodol droplets was also observed in liver and bladder. These results indicated the metabolization of Janus particles mainly proceeded with liver and kidney.

Supplementary Fig 29. TEM images of the Janus particles in the collected urine of rabbit post-embolization. In the collected urine of rabbit, we also clearly observed the Janus particles.

Comment 5: The authors suggest that the 8spheres may not be as viscoelastic as compared to the newly fabricated droplets. Mechanical testing needs to be carried out for 8spheres to support the argument for comparison.

Response: Thanks very much for the reviewer's good comments. According to the reviewer's suggestion, we have added the mechanical modulus of 8spheres in the revised manuscript, as shown in Fig. 3a and 3c.

In Page 7, Line 167-171; Line 175-180

“As shown in Fig. 3a, the viscosity of the Janus particle-engineered structural lipiodol droplets and clinical 8spheres® beads significantly decreased with an increase in the shear rate, thereby exhibiting good shear-thinning behavior. The results suggested that the decreased viscosity of these Janus particle-engineered structural lipiodol droplets and clinical 8spheres® beads upon shearing was beneficial to delivery.

In addition, the scanning oscillation amplitude results showed that the maximum storage modulus (G') of the Janus particle-engineered structural lipiodol droplets was approximately 710 Pa, while the maximum G' of clinical 8spheres® beads was approximately 3900 Pa (Fig. 3c). The results indicated that Janus particle-engineered structural lipiodol droplets exhibited much more viscoelastic deformation capacities

than the clinical 8spheres® beads.”

Fig. 3 | Viscoelasticity and embolization effectiveness of Janus particle-engineered structural lipiodol droplets in vitro. **a** Rheology of the Janus particle-engineered structural lipiodol droplets, lipiodol and 8spheres® beads by shear rate sweeps. The viscosity of the Janus particle-engineered structural lipiodol droplets and the 8spheres® beads significantly decreased with the increase of the shear rate, which is beneficial to transcatheter injection. **c** Storage modulus (G') and loss modulus (G'') of the Janus particle-engineered structural lipiodol droplets, lipiodol and 8spheres® beads as characterized by oscillatory strain sweeps, respectively. The results indicated that Janus particle-engineered structural lipiodol droplets exhibited much more viscoelastic deformation capacities than the clinical 8spheres® beads. In contrast, the clinical lipiodol exhibited liquid characteristics.

Comment 6: Have the authors look into additional tissue response besides H&E? Any inflammation or fibrosis in the microenvironment?

Response: Thanks very much for the reviewer’s good comments. According to the reviewer’s suggestion, we have looked into additional tissue response in the embolized right kidney and other organs as well as inflammatory factors in the blood of rabbits post-embolization. The results indicated that inflammation response and fibrosis in the microenvironment could be clearly observed in the Janus particle-engineered structural lipiodol droplets group and 8spheres® bead group. In particular, in the Janus particle-engineered structural lipiodol droplets group, an obvious infarction-triggered

inflammation zone could be observed, into which a large number of immune cells were recruited. Moreover, we also investigated other major organs (heart, liver, spleen, lung and left kidney) post-embolization in the Janus particle-engineered lipiodol droplet group. The results suggested that there was no obvious tissue damage and any inflammation or fibrosis in these organs. Meanwhile, we measured the inflammatory factors in the blood of rabbits post-embolization and the results suggested that the inflammatory factors in the blood of rabbit stayed within the normal range.

We have added the data and detailed discussions of the additional tissue response in the revised manuscript.

In Page 12, Line 294-303

“Besides, we also investigated additional tissue response in the embolized right kidney post-embolization. The results indicated that inflammation response and fibrosis in the microenvironment could be clearly observed in the Janus particle-engineered structural lipiodol droplets group and 8spheres® bead group (Supplementary Fig. 20). In particular, in the Janus particle-engineered structural lipiodol droplets group, an obvious infarction-triggered inflammation zone could be observed, into which a large number of immune cells were recruited. Moreover, we also investigated other major organs (heart, liver, spleen, lung and left kidney) post-embolization in the Janus particle-engineered lipiodol droplet group. The results suggested that there was no obvious tissue damage and any inflammation or fibrosis in these organs (Supplementary Fig. 21).”

In Page 11, Line 280-281; Page 12, Line 282-283

“We also measured the inflammatory factors in the blood of rabbit post-embolization and the results suggested that the inflammatory factors in the blood of rabbit stayed within the normal range (Supplementary Fig. 19).”

Supplementary Fig 20. H&E staining of the embolized right kidney post-embolization to look into inflammation response and fibrosis in the microenvironment. Scale bars are 2000 μm , 100 μm , 50 μm , 100 μm and 50 μm from left to right.

Supplementary Fig 21. H&E staining of major organs harvested from rabbit on days 45 post-embolization. Scale bars: 50 μm .

Supplementary Fig 19. The effect of Janus particle-engineered structural lipiodol droplets on sentinel pro-inflammatory in the blood of rabbit post-embolization.

Comment 7: The supplementary video description (in SI) does not match the video content. There is no in vitro retrieval. For the two videos, both appear to have materials leaking out, causing distal dislodgement that can be dangerous.

Response: Thanks very much for the reviewer's good comments. We apologize for our inaccurate descriptions. We have revised the video description in the revised supplementary information.

Regarding the material leaking, we have consulted the issue with clinicians. In the time of fourth seconds in Video S1, it is not the lipiodol leakage, but the renal excretion of contrast agent. This is a common phenomenon in renal artery angiography and is a manifestation of renal excretion of contrast agents. Before embolization, we injected 3 mL of contrast agent (iohexol) to confirm the morphology of the renal artery and the catheter location. The contrast agent (iohexol) would be excreted from kidney to downstream ureter. As shown in the following Figure, before injection of the Janus particle-engineered structural lipiodol droplets, renal excretion of contrast agent could also be observed.

Comment 8: There is no hemocompatibility testing. The interaction of any newly developed embolic material and blood is critical to investigate.

Response: Thanks very much for the reviewer's good comments. According to the reviewer's suggestion, we have supplemented the hemocompatibility testing of the Janus particle-engineered structural lipiodol droplets with hemolysis assays. The results indicated that the overall hemolysis rate of the Janus particle-engineered structural lipiodol droplets with different sizes was less than 2.5%, suggesting that the Janus particle-engineered structural lipiodol droplets could not trigger hemolysis. We have

added the results and discussions in the revised manuscript.

In Page 6, Line 150-155; Page 7, Line 156-157

“Hemocompatibility is a critical characteristic to investigate any newly developed embolic materials in directly contacting with blood. We have investigated the hemocompatibility testing of the Janus particle-engineered structural lipiodol droplets by hemolysis assays. The results indicated that the overall hemolysis rate of the Janus particle-engineered structural lipiodol droplets with different sizes was less than 2.5% (a hemolysis rate less than 5% is considered permissible), suggesting that the Janus particle-engineered structural lipiodol droplets could not trigger hemolysis (Supplementary Fig 11).”

Supplementary Fig 11. Hemolysis rate of the Janus particle-engineered structural lipiodol droplets with different sizes.

Response to Reviewer 3:

The authors presented the Janus Particle-Engineered Structural Lipiodol Droplets for Arterial Embolization. In the first part of the paper, the authors employed their previous synthetic technique to prepare the Janus particles.

Further, they used the prepared Janus particles for arterial embolization (liver and kidneys). The embolization results are quite promising as some other studies, but the synthesis mechanism and justification (structure-property relationship) show limited novelty.

Response: Thanks very much for the reviewer's highly positive comments on the embolization results of the Janus particle-engineered structural lipiodol droplets.

We would like to clarify the novelty of our manuscript. The novelty in this manuscript is the viscoelastic structural lipiodol droplets that enable them to adapt to different levels of blood vessels to persistently embolize the feeding arteries and travel distally to embolize finer vasculatures. Compared to the clinical lipiodol system, the self-assembly of Janus particles at the lipiodol-water interface would endow the obtained structural lipiodol droplets with good mechanical stability, affording them to closely pack and embolize the feeding artery. Compared to the solid beads, the self-assembly of Janus particles at the lipiodol-water interface may also endow the obtained structural lipiodol droplets with good viscoelasticity, making them travel distally to embolize finer vasculatures of artery by deformation. As indicated by our results, the highly efficient embolization can be achieved by the viscoelastic Janus particle-engineered structural lipiodol droplets without any evidence of recanalization and non-target embolization.

Since 1978, Prof. Yamada propose the transcatheter arterial chemoembolization technique, lipiodol embolization have been widely used in clinic due to the excellent capacity of radiography and broad drug loading in comparison with other embolic materials. However, lipiodol embolization system has been confirmed to be highly unstable and remains limited by the recanalization and non-target embolization. More importantly, most of the existing embolic materials of chemoembolization, such as those solid beads, are difficult to adapt to different levels of the blood vessels due to their poor viscoelastic deformation capacities as a result of the large sizes and rigid structures. We hope our attempts would improve the issues.

We have attached another manuscript for reviewing purposes about structural droplets dominated by interfacial self-assembly of topology-tunable Janus particles towards the fabrication of macroscopic materials. In the attached manuscript we have demonstrated the structure-property relationship for the fabrication of Janus particle-engineered structural droplets.

Comment 2: The self-assembly of macro/microparticles is easy and has much literature available, but the self-assembly of Janus NPs at the nanoscale is very challenging.

Response: Thanks very much for the reviewer's good comments. We agree with the reviewer's viewpoints that the self-assembly of Janus nanoparticles at the nanoscale is challenging. In our manuscript, we focus the manuscript on arterial embolization by using several hundred micro-sized Janus particle-engineered structural lipiodol droplets. In clinical practice, such as transcatheter arterial chemoembolization, the embolic materials are usually needed several hundred micros (usually more than 100 μm). To meet the clinical needs, the self-assembly of Janus particles should be at the macro-scale. Thus, we prepared several hundred micro-sized Janus particle-engineered structural lipiodol droplets by the self-assembly of Janus particles.

To our best knowledge, there are two challenges in the self-assembly of nano/micron-sized particles to fabricate structural droplets. One great challenge is the self-assembly of Janus nanoparticles at the nanoscale, as mentioned by the Reviewer. And another challenge is the self-assembly of particles to fabricate several hundred micro-sized droplets because it needs great efforts to prevent the coalescence of these large-sized droplets. In our study, several hundred micro-sized structural lipiodol droplets stabilized by our Janus particles show considerable interfacial stability, allowing to effectively prevent their coalescences. The good stability and viscoelastic deformation of Janus particle-engineered structural lipiodol droplets afford them to achieve efficient embolization of arterial vasculatures

Comment 3: Moreover, characterization of Janus is also limited in this study, for example, SEM did not show any Janus structure, but it's quite easy to obtain SEM of Janus microparticles, also, the degree of Janus formation should be explained clearly.

Response: Thanks very much for the reviewer's comments. The synthesis and characterization of Janus particles have been reported in our previous works, including SEM and TEM characterization of Janus structure, the degree of the time-dependent growth process of Janus particles (Sci. Adv. 2017;3: e1603203; Macromolecules, 2018, 51, 1591; Macromolecules 2019, 52, 3237).

Fig. 2 From Science Advances paper. Emulsion interfacial polymerization mechanism for producing Janus particles. (a) Schematic representation of the fabrication of Janus particles. (b) Bright-field microscope images of the time-dependent growth process of Janus particles. Scale bars, 5 mm. (c) The fluorescent microscope images of the time-dependent growth process of Janus particles. Scale bars, 5 mm. (d) Computer simulation results. A dissipative dynamics simulation model combined with the stochastic reaction model is constructed to investigate the dynamic behavior during interfacial polymerization. The yellow bead represents the St and DVB, while the blue sticks represent the polymerized AA monomers.

Fig. 1 From Science Advances paper. **Synthesis and characterization of Janus particles.** (a) Highly-yield synthesis of typical crescent-moon shaped poly(styrene-co-divinyl benzene) \supset Poly(acrylic acid) (PSDVB \supset PAA) Janus particles. The resulting crescent-moon-shaped Janus particles exhibit uniform size and distribution. Scale bar 10 μm . Energy dispersive X-ray spectroscopy (EDX) analysis indicates the different O element distributions on their convex surfaces and concave surfaces of Janus particles. The O element only exists in PAA, implying the convex surface of the particle is hydrophilic PAA while the concave surface is hydrophobic PSDVB. (b) HRTEM image of embedded Janus particles after microtome cutting and staining with phosphotungstic acid. Scale bar 500 nm. (c) SEM images of topologically anisotropic Janus particles. The topological geometries of Janus particles can be well tuned from bread, hemisphere, crescent-moon to pistachio by regulating the concentrations of anchoring monomer. Scale bars, 2 μm . (d) A ternary phase-like diagram of AA, St and DVB used in the production of topologically anisotropic Janus particles.

Comment 4: In addition, the structural properties are not consistent in the capillary glass tube, and the uniformity of the structure is limited at a large scale.

Response: Thanks very much for the reviewer's comments. We apologize for our unclear descriptions. The inconsistent structural properties in the capillary glass tube should be attributed to the difference in packing density from the left side to the right side during injecting. In our study, the right side of the capillary glass tube was sealed. The Janus particle-engineered structural lipiodol droplets were injected into a semi-closed glass capillary tube from the left side to the right side at a constant speed of 0.5 mL/min. When those Janus particle-engineered structural lipiodol droplets preferentially reached the end of the tube (right side), they began to pack together tightly. With the constant packing, some of them were able to further deform into different shapes under continuous injection force. Thus, it resulted in inconsistent structural properties in the different zones of the capillary glass tube. The inconsistent structures of Janus particle-engineered structural lipiodol droplets can also be found in the results of *in vitro* embolization (Fig. 3k-o) and *in vivo* embolization (Fig. 5b, Supplementary Fig. 23). We have revised the descriptions in the revised manuscript.

In Page 8, Line 187-194

“Next, a glass capillary tube was used to visually mimic the embolization of the obtained Janus particle-engineered structural lipiodol droplets. The right side of the capillary glass tube was sealed. The Janus particle-engineered structural lipiodol droplets were injected from the left side to the right side of the capillary glass tube at a constant speed of 0.5 mL/min. When those Janus particle-engineered structural lipiodol droplets preferentially reached the end of the tube (right side), they began to pack together tightly. Some of them that approached the right side of the capillary glass tube were deformed into different shapes under continuous injection force.”

Comment 5: The ratio of cisplatin encapsulation needed to be explained (No of Particles per concentration of cisplatin). The above mentioned concerns showed limited novelty to be published in this journal.

Response: Thanks very much for the reviewer's comments. Cisplatin was encapsulated in the oily core of the Janus particle-engineered structural lipiodol droplets. According to Reviewer's suggestion, we have calculated that 1 mL lipiodol can produce

approximately 6000 Janus particle-engineered structural lipiodol droplets. For example, in a typical encapsulation, the encapsulation efficiency of cisplatin within Janus particle-engineered structural lipiodol droplets (containing 0.1 mL of lipiodol, approximately 600 droplets) was approximately 93% at the feed amount of 6 mg. Thus, 1 mg of cisplatin could be loaded into approximately 107 Janus particle-engineered structural lipiodol droplets. We have added the information in the revised manuscript.

In Page 5, Line 105-106

“We have calculated that 1 mL lipiodol can produce 6000 Janus particle-engineered structural lipiodol droplets ($120 \pm 40 \mu\text{m}$).”

In Page 6, Line 133-135

“In other words, for example, in a typical encapsulation of cisplatin at 6 mg, 1 mg of cisplatin could be loaded into approximately 107 Janus particle-engineered structural lipiodol droplets in the case.”

Reviewers' Comments:

Reviewer #1:

Remarks to the Author:

I carefully studied the reply of the authors to the comments I gave previously. The authors modified the text accordingly. I would like to thank them for all the comments that they made. Therefore, I do think it can be accepted for publication.

Reviewer #2:

Remarks to the Author:

The reviewer appreciates the efforts the authors made. All comments have been addressed properly.

Reviewer #3:

Remarks to the Author:

The revised manuscript presented satisfactory results.

List of Responses to the Reviews' Comments

Thanks a lot for the reviewers' comments.

Response to Reviewer 1:

I carefully studied the reply of the authors to the comments I gave previously. The authors modified the text accordingly. I would like to thank them for all the comments that they made. Therefore, I do think it can be accepted for publication.

Response: We thank the reviewer's highly positive comments and insightful suggestions to improve the quality of our manuscript.

Response to Reviewer 2:

The reviewer appreciates the efforts the authors made. All comments have been addressed properly.

Response: Thanks very much for the reviewer's highly positive comments and helpful suggestions.

Response to Reviewer 3:

The revised manuscript presented satisfactory results.

Response: We thank the reviewer's invaluable comments and feedback.